# MVA: Linear Attention with High-order Query-Keys Integration and Multi-level Vocabulary Decomposition

**Ning Wang** [* 1 2 3]  **Zekun Li** [* 1 2 3]  **Tongxin Bai** [3]  **Man Yao** [1 2]  **Zhen Qin** [4]  **Guoqi Li** [1 2]

## Abstract

Linear attention offers the advantages of linear inference time and fixed memory usage compared to Softmax attention. However, training large-scale language models with linear attention from scratch remains prohibitively expensive and exhibits significant performance gaps compared to Softmax-based models. To address these challenges, we focus on transforming pre-trained Softmax-based language models into linear attention models. We unify mainstream linear attention methods using a **high-order QK integration theory** and a **multi-level vocabulary decomposition**. Specifically, the QK integration theory explains the efficacy of combining linear and sparse attention from the perspective of information collection across different frequency bands. The multi-level vocabulary decomposition exponentially expands memory capacity by recursively exploiting compression loss from compressed states. To further improve performance and reduce training costs, we adopt a **soft integration strategy** with attention scores, effectively combining a sliding window mechanism. With less than 100M tokens, our method fine-tunes models to achieve linear complexity while retaining 99% of their original performance. Compared to state-of-the-art linear attention model and method, our approach improves MMLU scores by 1.2 percentage points with minimal fine-tuning. Furthermore, even without the sliding window mechanism, our method achieves state-of-the-art performance on all test sets with 10B tokens.

---
[*]Equal contribution [1]Institute of Automation, Chinese Academy of Sciences [2]School of Artificial Intelligence, University of Chinese Academy of Sciences [3]Beijing Academy of Artificial Intelligence [4]TapTap. Correspondence to: Tongxin Bai <txbai@baai.ac.cn>, Guoqi Li <guoqi.li@ia.ac.cn>.

*Proceedings of the 42nd International Conference on Machine Learning*, Vancouver, Canada. PMLR 267, 2025. Copyright 2025 by the author(s).

## 1. Introduction

The attention mechanism has been the cornerstone of recent advancements in natural language processing, computer vision, and related fields. Current large language models (LLMs) (Touvron et al., 2023; Jiang et al., 2023) predominantly leverage the Transformer architecture (Vaswani et al., 2017) with Softmax Attention, demonstrating remarkable performance. However, the Softmax Attention mechanism inherently suffers from quadratic complexity in both time and space during training. This issue is particularly pronounced during inference, where the KV cache grows linearly with sequence length, leading to substantial computational costs and memory overhead. These constraints significantly hinder efficient deployment in real-world applications.

To address this bottleneck, various linearized recursive models (Zaheer et al., 2020; Katharopoulos et al., 2020; Chou et al., 2024; Zhang et al., 2024b) have been proposed. These approaches aim to reduce the complexity of self-attention from quadratic to linear while keeping the KV cache size constant. This enables more efficient inference with fewer computational and memory resources. However, training a linear model from scratch requires substantial resources and faces numerous challenges. Consequently, a widely accepted and researched approach is to repurpose the pre-trained weights of Softmax Attention-based LLMs into linear-complexity language models by fine-tuning (Kasai et al., 2021). This strategy not only conserves significant resources but also enables linear models to approach, and in some tasks surpass, the performance of their Softmax-based counterparts. The key to achieving this transformation lies in approximating Softmax Attention with minimal cost, either through functional or computational approximations.

Traditional linear attention models and state-space models (SSMs) without gating mechanism, such as Linformer (Wang et al., 2020a), Performer (Choromanski et al., 2022), S4 (Gu et al., 2022), RetNet (Sun et al., 2023), and H3 (Fu et al., 2023), exhibit significant discrepancies from Softmax Attention due to their inability to functionally approximate the Softmax operation. This limitation makes it challenging to inherit pretrained weights. Recent research has introduced mechanisms such as gating to ap-

proximate the functionality of Softmax Attention, leading to models like HGRN (Qin et al., 2023), Gated Linear Attention (GLA) (Yang et al., 2024b), Gated Slot Attention (GSA) (Zhang et al., 2024b), Unified Optimal Linear Attention (MetaLA) (Chou et al., 2024), and Mamba (Gu & Dao, 2024). Among these, MetaLA provides a unified framework for analyzing existing linear models and demonstrates their potential for optimal approximation of Softmax Attention. However, we observe that the model described above lacks the Softmax function's ability to capture both high-frequency and low-frequency information. By analyzing the errors (appendix C) between Softmax Attention, GSA, and MetaLA, we find that GSA, due to its query-based compression of the key-value (KV) sequences, tends to focus on high-frequency information, or potentially introducing high-frequency noise. In contrast, the MetaLA family of linear attention methods primarily captures the low-frequency components of the original attention map. In addition, none of these methods efficiently enhance memory capacity, which could help reduce the error between linear models and Softmax Attention, which has infinite cache expansion.

Based on the above analysis, we propose three steps to continually reduce the error with Softmax Attention: (1) First, we propose using recursive sparse attention approach to focus on high-frequency information fitting, while employing unified optimal linear attention to dominate the fitting of low-frequency information and suppress the high-frequency noise introduced by recursive sparse attention. In this way, we achieve a unification and combination of recursive sparse attention and unified optimal linear attention. We refer to this approach of progressively combining different frequency bands as the **Higher-Order QK Integration Theory**. (2) After unifying the two most effective methods, we propose a unified approach for efficient memory capacity expansion: the **Multi-Level Vocabulary Decomposition Method**. All recursive attention mechanisms with linear complexity compress the infinitely growing KV sequences into fixed-size states, which inherently leads to information loss and error. By recursively compressing this error into a series of states, we achieve an exponential decrease in the expected error as the level increases. This results in a polynomial increase in memory capacity, significantly reducing the error compared to Softmax Attention. Compared to the Delta Rule, we store the Delta separately rather than directly adding it to the original state. We name the resulting attention mechanism **MVA**.

Additionally, recent hybrid architectures such as Based (Arora et al., 2024), LoLCATs (Zhang et al., 2024a), and Distill to Mamba (Wang et al., 2025) offer promising solutions by combining sliding window attention (SWA) with linear attention or replacing specific layers of SSMs with attention mechanisms, which can significantly

restore LLM performance and save a lot of resources. However, these approaches often suffer from suboptimal convergence due to the significant gap between their linear models and Softmax Attention, and overfitting within the window. (3) Thus, we introduced SWA and solved the above problem. By maintaining both the key ($K$) and value ($V$) states, we retain the attention score during computation. This enables us to balance the historical information encoded by MVA with the current information from the SWA, named MVA-SW, achieving faster convergence, reduced resource requirements, and superior performance.

Using these methods, our linear model restores 99% of Mistral-7B's performance with fewer than 100M fine-tuning tokens. Compared to GSA, our MVA without using the sliding window outperforms it across all test sets using only half the training tokens.

## 2. Background and Preliminaries

### 2.1. Transformer

Transformer leverages Softmax Attention with uncompressed Key-Value (KV) cache. For an input sequence $X \in \mathbb{R}^{n \times d}$, the attention computation is defined as follows:
**Parallel Form:**

$$O = \text{Attention}(Q, K, V) = \text{Softmax}\left(\frac{QK^\top}{\sqrt{d_k}} \odot M\right) V,$$

**Recursive Form:**

$$K_t = \text{concat}(K_{t-1}, k_t), \quad V_t = \text{concat}(V_{t-1}, v_t)$$

$$o_t = \text{Softmax}\left(\frac{q_t K_t^\top}{\sqrt{d_k}}\right) V_t$$

where $Q = XW_q$, $K = XW_k$, $V = XW_v$, with $Q, K \in \mathbb{R}^{n \times d_k}$ and $V \in \mathbb{R}^{n \times d_v}$. $M$ represents the mask, which is a causal mask for language models, i.e., a lower triangular matrix filled with ones below the diagonal and $-\infty$ elsewhere. In the recursive formulation, the subscript $t$ denotes the input required for attention at timestep $t$. Specifically, $K_t \in \mathbb{R}^{t \times d_k}$, $V_t \in \mathbb{R}^{t \times d_v}$, $q_t, k_t \in \mathbb{R}^{1 \times d_k}$, and $v_t, o_t \in \mathbb{R}^{1 \times d_v}$.

The space and time complexity of training this formulation is quadratic. FlashAttention(Dao, 2023) reduces the memory complexity to nearly linear, but during inference, the KV cache size still grows linearly with sequence length, leading to significant memory overhead. This can even result in memory explosion for long sequences, making tasks infeasible. Thus, various approaches, such as linear attention and state-space models (SSMs), have been developed to address this issue by maintaining a fixed-size state for inference.

### 2.2. Linear Attention

Linear Attention computes attention as follows:

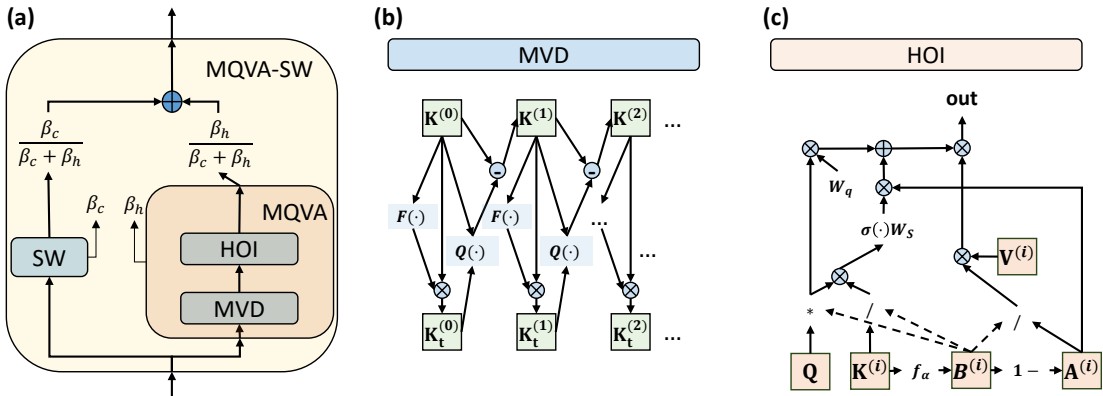

*Figure 1.* (a) The Attention mechanism based on High-order QK integration (HOI), multi-level vocabulary decomposition (MVA) and sliding window attention (SW). (b) The original K and V sequences are first decomposed to address the information loss caused by compressing them into fixed-size hidden states during recursion. By computing the difference between the original sequences and hidden states, we estimate the information error and further compress it to reduce this error. (c) is high-order QK integration Attention mechanism, where QK sequences of different orders are used for pseudo-querying and different frequencies information gathering. $*$ is element-wise multiplication, $/$ is element-wise division, $\otimes$ is matrix multiplication.

**Parallel Form:**

$$O = \text{LA}(\phi(Q), \phi(K), V) = \left( \left( \phi(Q)\phi(K)^\top \right) \odot M \right) V,$$

**Recursive Form:**

$$S_t = S_{t-1} + \phi(k_t)^\top v_t, \quad o_t = \phi(q_t)S_t,$$

where $S_t \in \mathbb{R}^{d_k \times d_v}$. Linear Attention maintains a fixed-size state $S_t$, achieving constant memory complexity during inference. However, it suffers from the issue of diluted attention, where the model fails to focus effectively on relevant tokens. Many works address this by improving the function $\phi$ (Han et al., 2023; Choromanski et al., 2022) or introducing additional mechanisms. Linear Attention also struggles to leverage positional biases (Su et al., 2023) that prioritize neighboring tokens and memory forgetting, leading to the development of Gated Linear Attention (GLA).

### 2.2.1. GATED LINEAR ATTENTION (GLA)

GLA (Yang et al., 2024b) introduces a gating mechanism to alleviate the limitations of Linear Attention:

**Parallel Form:**

$$O = \text{GLA}(Q, K, V, G) = \text{LA}(Q \odot B, \frac{K}{B}, V), \quad (1)$$

where the $t$-th row of $B$ is defined as $b_t = \prod_{i=1}^{t} g_i$, with $g_i$ being the $i$-th row of the gating matrix $G = \sigma(XW_g) \in \mathbb{R}^{n \times d_k}$.

**Recursive Form:**

$$S_t = \text{diag}(g_t)S_{t-1} + k_t^\top v_t, \quad o_t = q_t S_t.$$

The gating mechanism provides input-dependent dynamic decay, emphasizing neighboring tokens and adjusting the decay size per step. This addresses attention dilution and improves the state update process with forgetting unimportant information.

### 2.2.2. META LINEAR ATTENTION (METALA)

MetaLA (Chou et al., 2024) unifies and optimizes linear attention with the following formulation:

**Parallel Form:**

$$O = \text{MetaLA}(Q, V, G) = \text{LA}(Q \odot B, \frac{1-G}{B}, V), \quad (2)$$

where $B$ and $G$ are defined as in GLA.

**Recursive Form:**

$$S_t = \text{diag}(g_t)S_{t-1} + (1-g_t)^\top v_t, \quad o_t = q_t S_t.$$

MetaLA uses fewer parameters while achieving superior performance. It can converge to existing Linear Attention, LinRNN, and SSM models under specific conditions. However, as MetaLA removes the $K$ matrix, it introduces a significant gap from Softmax Attention, hindering methods that rely on fine-tuning Softmax Attention weights. We propose a solution to bridge this gap while unifying recursive sparse attention and delta-rule-based mechanisms.

### 2.3. Recursive Sparse Attention

Recursive sparse attention compresses $K$ and $V$ sequences incrementally. For example, Gated Slot Attention (GSA) (Zhang et al., 2024b) is formulated as:

**Recursive Form:**

$$K_t = \text{diag}(g_t)K_{t-1} + (1-g_t)^\top k_t, \quad V_t = \text{diag}(g_t)V_{t-1} + (1-g_t)^\top v_t$$

$$o_t = \text{Softmax}(q_t K_t^\top)V_t.$$

**Parallel Form:**

$$O = \text{GSA}(Q, K, V, G) = \text{GLA}(\text{Softmax}(\text{GLA}^\top(Q, K, 1-G, G)), 1-G, V, G). \tag{3}$$

GSA retains the $K$ and $V$ sequences, offering advantages for linearizing Softmax Attention while preserving its weights. Experimental results indicate that replacing Softmax with a perceptron (e.g., $\sigma(\cdot)W$) leads to faster convergence and better performance. Our method also preserves $K$ and $V$ sequences while employing a perceptron to replace Softmax, further incorporating a delta-rule-inspired mechanism to expand memory capacity exponentially.

### 2.4. Delta Rule Memory Mechanism

Delta rule-based (Schlag et al., 2021) updates refine state transitions by querying existing states before updates:

**Recursive Form:**

$$v_t^{\text{old}} = k_t S_{t-1}, \quad S_t = S_{t-1} + g_t \cdot k_t^\top (v_t - v_t^{\text{old}}), \quad o_t = q_t S_t.$$

The parallel form is detailed in related works. The delta rule updates states by incorporating only the new information, avoiding redundant accumulation.

## 3. Methodology

In this study, we propose two theoretical frameworks to effectively integrate and extend existing linear attention mechanisms (e.g., MetaLA) and recursive compression attention mechanisms (e.g., Gated Slot Attention, GSA).

The first framework, termed *High-order QK Integration Theory*, introduces pseudo-queries of different orders, enabling information collection at varying frequencies while efficiently storing the retrieved information in the $KV$ cache. This theoretical foundation provides a unified perspective for integrating linear attention with recursive compression attention. Furthermore, inspired by the Delta Rule and the vocabulary decomposition techniques, we propose a **Multi-level Quantized Vocabulary Decomposition** method to efficiently enhance memory capacity.

To further substantiate our approach, we conduct an error analysis comparing our method with full attention. The results demonstrate that the proposed techniques enable a more accurate approximation to full attention compared to existing methods. This improved approximation is particularly beneficial for the implementation of hybrid attention

or hybrid architectures. Therefore, we introduce a hybrid attention mechanism with **soft integration by attention scores** that combines sliding window for processing current information and linear attention for handling historical information. By leveraging their respective attention scores, our method effectively balances contributions from historical and current information, further improving model performance on various tasks.

### 3.1. High-order QK Integration

In this section, we introduce the High-order $QK$ Integration Theory to unify and extend linear attention mechanisms, such as MetaLA, and recursively compressed attention mechanisms, such as GSA. We begin by examining the original self-attention mechanism in full attention: $e^{QK^\top}$.

**Theorem 3.1.** . *Given a self-attention mechanism expressed as $e^{QK^\top}$, the Taylor series expansion of the $(i, j)$-th element is given by:*

$$
\begin{aligned}
e^{q_i k_j^\top} &= 1 + \sum_m \frac{1}{m!}(q_i k_j^\top)^m = \sum_{m=0}^\infty \sum_{s_1+s_2+\cdots+s_d=m} \prod_{l=1}^d \frac{(q_{il}k_{jl})^{s_l}}{s_l!} \\
&= \sum_{s_1,\ldots,s_d} F_{ms_l}(q_{il}) \prod_{l=1}^d (k_{jl})^{s_l} \\
&= \sum_{s_1,\ldots,s_d} F_{ms_l}(q_i) k_j^{\top s_1} A_{ms_1} k_j^{\top s_2} \ldots A_{ms_{d-1}} k_j^{s_{d-1}} A_{ms_d} k_j^{\top s_d}
\end{aligned}
$$

*where $A_{mu} \in \mathbb{R}^{d_k \times d_k}$ is a matrix independent of $q$ and $k$ and $F_{ms_l}(q_i) = q_i^{s_1} A_{ms_1} q_i^{\top s_2} \ldots A_{ms_d} q_i^{\top s_d}$.* □

This expansion reveals that attention scores can be expressed as higher-order terms of $q$ and $k$, where the different powers of $k$ control the contributions from various frequency components. This observation is central to our high-order integration framework, where we explore how different frequencies of information can be selectively integrated.

Next, we investigate the ability of linear models such as MetaLA and recursively compressed attention models such as GSA to represent different types of information.

**Theorem 3.2** (High-order $QK$ Integration). *By independently setting different frequency components for queries $(Q)$ and keys $(K)$, linear models (e.g., MetaLA) use low-frequency fixed polynomial $Q$ queries over multi-frequency $K$ polynomials, while recursive models (e.g., GSA) employ varying-frequency polynomial $Q$ queries over fixed $K$ polynomials. This makes GSA more sensitive to high-frequency information. Thus, MetaLA's low-frequency focus complements GSA's high-frequency sensitivity, enabling a more balanced and effective attention mechanism.*

**Proof.** We analyze the attention scores, temporarily neglecting the impact of $B_n$ or treating $B_n$ as learnable positional encodings and dynamic decay.

1. The calculation of linear attention can be expressed as:

$$\phi(q_n)\mathrm{F}(CK_n^\top)V_n.$$

If $F$ is the identity function, the method reduces to Linear Attention (LA). When $F$ takes a non-linear form involving an exponential, such as $\frac{e^{-x}}{1+e^{-x}}$ in MetaLA, it enhances the first-order information by propagating it to all powers of $K$ via a fixed non-linear function.

Neglecting the denominator for simplicity, as its primary role is normalization (leaving more detailed analysis for future work). The $(i,j)$-th attention score of MetaLA is:

$$\sum_{t=0} \frac{1}{t!}\phi(q_i)(Ck_j^\top)^t = \sum_{s_1,s_2,\dots,s_d}\left(\sum_t \phi(q_{it})\mathrm{F}_{ms_l}(C_{tl})\right)\prod_{l=1}^d (k_{jl})^s$$

where $\phi(x) = xW$, the parameter $W$ ensures that the shape of $q$ is $1 \times m$. This formulation enables fixed $Q$ to query each frequency of $K$ independently, effectively eliminating first-order approximation errors. By carefully designing $\phi(\cdot)$ as a polynomial function, higher-order errors can also be mitigated. In addition, causal convolutions in these methods can be interpreted as zero-order integration.

2. Similarly, the Gated Slot Attention (GSA) mechanism can be formulated as:

$$\mathrm{Softmax}(q_n(\mathrm{F}(CK_n^\top)K_n)^\top)\mathrm{F}(CK_n^\top)V_n,$$

where $\phi(q_n)$ is replaced by $\mathrm{Softmax}(q_n(\mathrm{F}(CK_n^\top)K_n)^\top)$. Considering the numerator, while mitigating the impact of denominator by a final linear mapping, the expanded form becomes, results in the expanded form:

$$\sum_{s_1,s_2,\dots,s_d}\left(\sum_t\sum_u \frac{1}{u!}\left(q_i\sum_{a=0}^i k_a^\top \mathrm{F}_t(k_aC^\top)\right)^u \mathrm{F}_{s_l}(C_{tl})\right)\prod_{l=1}^d (k_{jl})^s$$

By approximating $\sum_{a=0}^i k_a^\top \mathrm{F}_t(k_aC^\top)$ with an average matrix $\bar{C}_a$, this can be further rewritten as:

$$\sum_{s_1,s_2,\dots,s_d}\left(\sum_t\sum_{u_1,u_2,\dots,u_d} \mathrm{F}_{u_l}(q_i)\prod_{l=1}^d (\bar{C}_{at})^u \right)\mathrm{F}_{s_l}(C_{tl})\prod_{l=1}^d (k_{jl})^s.$$

Since $s$ and $u$ are at least 1, and $\bar{C}_j$ represents at least second-order $k_j$ terms multiplied by $k_j$ itself, GSA primarily emphasizes high-frequency information from higher-order terms, while lacking adequate contributions from low-frequency components. By integrating MetaLA, which complements these missing low-frequency details, a more accurate approximation can be achieved, significantly reducing resource consumption and improving performance. For detailed error analysis of GSA and its complementarity with MetaLA, please refer to the appendix C.1. $\qquad\square$

## 3.2. Multi-level Vocabulary Decomposition

In recursive linear models, the challenge lies in efficiently compressing sequences that grow linearly into compact hidden states. These models aim to retain critical information while minimizing memory requirements. Specifically, given the compressed states $K_t = \mathrm{F}(CK^\top)K \in \mathbb{R}^{m\times d_k}$ and $V_t \in \mathbb{R}^{m\times d_v}$, some information from the original sequence is inevitably lost during the compression process. Here, $C \in \mathbb{R}^{m\times d_k}$ is a learnable matrix and $K \in \mathbb{R}^{t\times d_k}$ represents the sequence of keys. The error introduced by this compression can be expressed as:

$$\epsilon^{(i)} = K^{(i)} - S(K^{(i)}K_t^{(i)\top})K_t^{(i)}, \quad K^{(i+1)} = \epsilon^{(i)}W^{(i)}, \quad (4)$$

where $\mathrm{S}(A)$ is defined as:

$$\mathrm{S}(A) = \begin{cases} 1 & \text{if } A_{ij} \text{ is the maximum in row i} \\ 0 & \text{otherwise} \end{cases}.$$

The lost information is compressed into $K_t^{(1)}$, where $K^{(0)}$ is the original $K$. Iteratively applying this process, the error expectation can decay at an exponential rate, by yielding a series of states $K_t^{(i)}$ ($i \leq h$, where $h$ is the preset number of levels). Assuming a $c$-level recursion, the final error is given by:

$$E = K^{(c+1)} = (I - L_c)K^{(c)}W_c = \prod_{i=0}^c (I - L_i)K\prod_{i=0}^c W_i.$$

$$(5)$$

**Theorem 3.3** (Improved Memory Capacity). *After $c$ recursions, the expected error $E$ can be bounded as follows:*

$$\|E\| \leq \|K\|\prod_{i=0}^c \epsilon_i \prod_{i=0}^c \gamma_i,$$

*where $\epsilon_i = \frac{n-m}{n}$ and $\gamma_i$ is determined by learnable parameters that ensure convergence to a value less than 1.*

The analysis and proof are given in Appendix C.2. $\qquad\square$

Theorem 3.3 demonstrates that the error can decrease at an exponential rate. Conversely, it also implies that the memory capacity can scale exponentially.

The $S$-function in Equation 4 requires the use of the `gather` operation. However, we consider replacing it with a more efficient Softmax approximation. The term $\mathrm{S}(K^{(i)}K_t^{(i)\top})K_t^{(i)}$ can be implemented using two rounds of the GLA or GSA operator in parallel. Furthermore, multi-level parallelism can be combined with Equation 5 for implementation. To simplify and improve efficiency, we observe that replacing it with $\mathrm{Softmax}(KC^\top)C$ or $\mathrm{Softmax}(KC^\top)K_t$ yields comparable performance. The latter can be implemented using $\mathrm{GLA}(\mathrm{Softmax}(KC^\top), 1 - G, K, G)$. Alternatively, a perceptron can approximate S instead of Softmax.

### 3.3. MVA

Using the theoretical foundations introduced above, we develop a high-order $QK$ Integration and multi-level vocabulary decomposition attention mechanism (MVA). In the experiment, we used a high-order QK integration method, where the low order is processed by a branch similar to MetaLA and the high order is dominated by a branch similar to GSA, so that information of all frequencies can be processed more reasonably. The computation process is detailed as follows:

$$
\begin{aligned}
K_t^{(i)} &= \mathrm{diag}\left(\mathrm{f}_g^{(i)}\left(k_t^{(i)}\right)\right) K_{t-1}^{(i)} + \left(1 - \mathrm{f}_\alpha^{(i)}\left(k_t^{(i)}\right)\right) k_t^{(i)} \\
V_t^{(i)} &= \mathrm{diag}\left(\mathrm{f}_g^{(i)}\left(k_t^{(i)}\right)\right) V_{t-1}^{(i)} + \left(1 - \mathrm{f}_\alpha^{(i)}\left(k_t^{(i)}\right)\right) v_t^{(i)} \\
v_t^{(i)} &= v_t^{(0)} = v_t, \quad k_t^{(i+1)} = k_t^{(i)} - \phi_k\left(k_t^{(i)} W_{kc}^{(i)}\right) K_t^{(i)} \\
P_{MetaLA}^{(i)} &= \phi_q(q_t W_{qc}^{(i)}), P_{GSA}^{(i)} = \sigma\left(q_t K_t^{(i)\top}\right) W_S^{(i)} \quad (6) \\
o_t &= \sum_i \mathrm{diag}(w^{(i)})(P_{metala}^{(i)} + P_{gsa}^{(i)}) V_t^{(i)} \\
\mathrm{f}_g^{(i)}\left(k_t^{(i)}\right) &= \mathrm{f}_\alpha^{(i)}\left(k_t^{(i)}\right) = \sigma\left(C^{(i)} k_t^{(i)\top}\right), \sigma = \tfrac{1}{1+e^{-x}} \\
w^{(i)} &= \tfrac{P^{(i)}}{\sum_i P^{(i)}}, \quad \phi_q(x) = x, \quad \phi_k(x) = x
\end{aligned}
$$

Where $C^{(i)} \in \mathbb{R}^{d_q \times d_v}$. For the explicit low-order branch, a weight matrix $W_{qc}^{(i)} \in \mathbb{R}^{d_q \times d_v}$ is applied to adjust its contribution. This branch can fully replicate the core operation of MetaLA. For the high-order branch, we utilize a layer of perceptron to replace the Softmax function in GSA and a weight matrix $W_S^{(i)} \in \mathbb{R}^{d_q \times d_v}$ to adjust attention scores and determine its contribution, equivalent to the role of GSA. For vocabulary decomposition, we adopt the second parallelizable method:

$$
\mathrm{Softmax}(K W_{kc}^{(i)}) K_t, \quad W_{kc}^{(i)} \in \mathbb{R}^{d_k \times d_k}. \quad (7)
$$

The parameters $W_{qc}^{(i)}, W_S^{(i)}, W_{kc}^{(i)}$ are shared across all heads, resulting in a total parameter count of $3 \times d \times d$, where $d$ denotes the head dimension. Future work will consider introducing per-head parameters.

To further simplify, we express this as a linear attention operator. Leveraging the GLA operator notation, the process is as follows:

$$
o_t = \mathrm{MVA}(q_t, k_t^{(0-m)}, v_t^{(0-m)}, C, W_S, W_{kc}, W_{qc}) : \quad (8)
$$
$$
q_{ks_t} = \mathrm{GLA}(q_t, k_t^{(i)}, (1 - \mathrm{f}_\alpha^{(i)}(k_t^{(i)})), \mathrm{f}_g^{(i)}(k_t^{(i)}))
$$
$$
p_{MetaLA_t} = \phi_q(q_t W_{qc}^{(i)}), p_{GSA_t} = \sigma(q_{ks_t}) W_S^{(i)}
$$
$$
o_t = \mathrm{GLA}(p_{metala_t} + p_{gsa_t}, (1 - \mathrm{f}_\alpha^{(i)}(k_t^{(i)})), v_t^{(i)}, \mathrm{f}_g^{(i)}(k_t^{(i)}))
$$
$$
k_t^{(i+1)} = k_t^{(i)} - \mathrm{GLA}(\phi_k(k_t^{(i)} W_{kc}^{(i)}), (1 - \mathrm{f}_\alpha^{(i)}(k_t^{(i)})), k_t^{(i)}, \mathrm{f}_g^{(i)}(k_t^{(i)}))
$$

Where i refers to the i-th level. Specific cases of MVA reduce to well-known methods:

$$
\mathrm{MetaLA} = \mathrm{MVA}(q_t, k_t^{(0)}, v_t^{(0)}, C^{(0)}, 0, 0, I), \quad (9)
$$
$$
\mathrm{GSA} = \mathrm{MVA}(q_t, k_t^{(0)}, v_t^{(0)}, C^{(0)}, W_S, 0, 0). \quad (10)
$$

Further equivalence demonstrations with additional methods are provided in the appendix.

### 3.4. Integrating MVA with Sliding Window (MVA-SW)

To further enhance the proposed MVA method, we incorporate the Sliding Window (SW) attention mechanism. The SW mechanism attention retains the distribution of the original attention, which allows us to significantly preserve performance while substantially reducing the training data required for convergence. Additionally, since our method reduces gap compared to Softmax, the historical attention scores retained are more consistent with the original attention. Balancing the outputs of MVA and SW using the current and historical attention scores is thus both natural and reasonable. The computation process is detailed as follows:

$$
\mathrm{out}_c = \mathrm{Softmax}(q_t[k_{(t-\mathrm{sw})} : k_t]^\top)[v_{(t-\mathrm{sw})} : v_t]
$$
$$
\mathrm{out}_h = (\mathrm{Softmax}(q_t K_t^\top) W_S + \phi(q_t)) \Lambda_t (1 - A_n) V_n
$$
$$
\mathrm{output} = \mathrm{diag}\left(\frac{\beta_c}{\beta_c + \beta_h}\right) \mathrm{out}_c + \mathrm{diag}\left(\frac{\beta_h}{\beta_c + \beta_h}\right) \mathrm{out}_h
$$

where

$$
\beta_c = \exp(q_t[k_{(t-\mathrm{sw})} : k_t]^\top) R_d, \quad (11)
$$
$$
\beta_h = \mathrm{GLA}(q_{ks_t} + \phi(q_t), 1 - G, I_V, G) W_{\beta_h} \quad (12)
$$

where $W_{\beta_h} \in \mathbb{R}^{d_v \times 1}$ and $R_d \in \mathbb{R}^{s_w \times 1}$. And, $\beta_h$ corresponds to a weighted summation over the attention map:

$$
((\mathrm{Softmax}(q_t K_t^\top) W_S + \phi(q_t)) \Lambda_t (1 - A_n)).
$$

The process of balancing historical and current information based on the attention scores closely aligns with the behavior of the original attention mechanism. This alignment further reduces the approximation error, ensuring a more faithful representation of the original attention while preserving computational efficiency.

## 4. Experiments

In this work, we explore inheriting LLM weights and converting them into linear models. Specifically, we adopt the Mistral-7B model as the base LLM and evaluate the performance of MVA-SW and MVA. We use the lm-evaluation-harness (Gao et al., 2024) tool to perform the test. For fine-tuning, we utilize LoRA (Hu et al., 2021) to achieve efficient fine-tuning, significantly reducing computational resources.

For MVA, we compare with the state-of-the-art GSA, as well as GLA, RetNet, and SUPRA (Mercat et al., 2024), which were benchmarked in the GSA paper. For MVA-SW, we

*Table 1.* MVA-SW fine-tuning results.

**Performance comparison across various 7B models. ♣ denotes models using softmax-attention. † denotes our results.**

| Shot(s) | Size | Tokens | ARC$_e$ 0 | ARC$_c$ 0 | Hella. 0 | PIQA 0 | Wino. 0 | NQ 5 | TriviaQA 5 | MMLU 5 | Avg. |
|---|---|---|---|---|---|---|---|---|---|---|---|
| *Models trained from scratch (for reference)* | | | | | | | | | | | |
| RWKV6 | 7B | 1.4T | 73.6 | 44.0 | 75.2 | 78.4 | 68.5 | 20.9 | 59.5 | 43.9 | 58.0 |
| Mamba | 7B | 1.2T | 77.6 | 46.8 | 77.8 | 81.0 | 72.3 | 25.4 | 66.2 | 33.2 | 60.0 |
| Llama2♣ | 7B | 2T | 76.4 | 46.2 | 76.0 | 78.0 | 69.2 | 26.0 | 64.2 | 45.5 | 60.2 |
| Gemma♣ | 7B | 6T | 81.5 | 53.2 | 80.5 | 79.8 | 74.0 | 24.3 | 63.7 | 63.2 | 65.0 |
| Mistral♣ | 7B | ? | 80.8 | 54.0 | 81.1 | 80.6 | 74.0 | 29.7 | 70.3 | 62.4 | 66.6 |
| *Models finetuned from Mistral 7B* | | | | | | | | | | | |
| SW-128† | 7B | **+0.1B** | 80.1 | 53.2 | 80.7 | 81.6 | 73.8 | 28.6 | 69.8 | 52.1 | 65.0 |
| GLA-SW-128† | 7B | **+0.1B** | 75.4 | 46.7 | 74.4 | 78.5 | 64.0 | 13.9 | 49.2 | 30.2 | 54.1 |
| GSA-SW-128† | 7B | **+0.1B** | 79.9 | 53.4 | 80.5 | 81.7 | 73.9 | 29.0 | 69.8 | 54.5 | 65.3 |
| MVA-SW(Ours)† | 7B | **+0.1B** | 80.5 | 54.3 | 80.8 | 82.0 | 74.0 | 29.6 | 70.1 | 57.1 | 66.1 |
| MVA-SW(Ours)† | 7B | **+2B** | 80.7 | 54.4 | 81.2 | 82.0 | 73.8 | 29.7 | 70.2 | 57.3 | 66.2 |

*Table 2.* Performance comparison between Qwen2.5 models and their MVA-SW converted versions on MMLU, PIQA, and Hellaswag benchmarks. Missing values are marked with "-".

| Model | MMLU | PIQA | Hellaswag |
|---|---|---|---|
| Qwen2.5-14B-1M | 80.7 | 85.2 | 87.3 |
| → MVA-SW (14B) | 77.3 | 83.8 | 86.8 |
| Qwen2.5-32B | 83.9 | - | 85.2 |
| → MVA-SW (32B) | 79.8 | 82.5 | 85.0 |

compare with LoLCATs, using identical fine-tuning parameters and window sizes for a fair comparison. In addition, we conduct experiments with GSA and GLA combined with the sliding window, using different fine-tuning parameter configurations for a comparative study with our method.

### 4.1. MVA-SW

This section evaluates MVA-SW with attention scores balancing approach. The integration follows the soft combination method described in Section 3.4, with a window size equal to the head dimension ($d_{head} = 128$). This ensures that the computational complexity of adding the sliding window remains equivalent to linear Attention ($O(nd^2)$) while preserving more of the model's performance. For fine-tuning, we use LoRA with the QKV mapping and FFN down_proj parameters, setting the rank to 128, alternatively, tuning only the QKV mapping with a rank of 8. Additionally, all parameters introduced by the linear attention part of MVA-SW are fine-tuned, while other parameters remain frozen. Optimization is performed using AdamW with a cosine learning rate schedule, an initial learning rate of $4 \times 10^{-5}$, 20 steps of linear warmup, and a training length of 1.5K due to GPU memory constraints, with a batch size of 0.1M tokens. The dataset used is the SlimPajama corpus. The results are shown in Table 1 and Table 2.

The results demonstrate that our method can recover most of the model's performance with a small number of tokens.

Furthermore, compared to GSA and GLA, our approach achieves better performance recovery due to its smaller error relative to Softmax Attention and the rationality of the soft combination mechanism. If continue training a large number of tokens we consider turning off the parameters of QKV and only consider the additional parameters introduced by training LA or truncating the reverse gradient information of SW and update only the information of LA. We froze the weights of QKV to train only the linear part to 2B tokens and its performance was slightly improved.

### 4.2. Comparison with LoLCATs

This experiment compares MVA-SW with LoLCATs. Unlike LoLCATs, which require a two-stage process, our method achieves comparable performance using a simple soft combination mechanism and straightforward fine-tuning. The experimental configuration matches LoLCATs, with a window size of 64. LoRA fine-tuning is applied to the QKV mapping parameters with a rank of 8. For MVA-specific parameters, full fine-tuning is applied, while other parameters remain frozen. The training length is 1K with a learning rate of $1 \times 10^{-4}$ and a batch size of 32K tokens. The results are presented in Table 3. In addition, we also investigated the effect of different fine-tuned datasets on MVA-SW and compared them with LoLCATs, as shown in Table 4. Our method requires only half the fine-tuning tokens to outperform LoLCATs on the MMLU tasks.

### 4.3. MVA

We replace the Attention mechanism in the Mistral model with MVA (MVA-SW without SWA), leaving the other components unchanged. The LoRA fine-tuning is applied to the QKV weights and the down_proj parameters in FFN, with a rank of 256. In addition, we unfreeze the embedding and normalization parameters while keeping all other parameters frozen. The optimization uses AdamW with a cosine learning rate schedule, an initial learning rate of $8 \times 10^{-5}$,

*Table 3.* Comparison between MVA-SW and LoLCATs.

| Model | Training Tokens (B) | PiQA | ARC-e | ARC-c (norm) | HellaSwag (norm) | Winogrande | MMLU (5-shot) | Avg. | Avg. (no MMLU) |
|---|---|---|---|---|---|---|---|---|---|
| Mistral 7B | - | 82.1 | 80.9 | 53.8 | 81.0 | 74.0 | 62.4 | 72.4 | 74.4 |
| Mistral 7B SUPRA | 100 | 80.4 | 75.9 | 45.8 | 77.1 | 70.3 | 34.2 | 64.0 | 69.9 |
| Mistral 7B LoLCATs | 0.04 | 81.5 | **81.7** | **54.9** | **80.7** | 74.0 | 51.4 | 70.7 | 74.5 |
| **Mistral 7B MVA-SW (Ours)** | **0.02** | **82.5** | 80.6 | 53.8 | 80.6 | **74.5** | 52.6 | **70.8** | 74.4 |

*Table 4.* Performance comparison on Alpaca-Clean and RedPajama datasets. All models are derived from Mistral-7B.

| Model | Training Data | PIQA | ARC-e | ARC-c | HellaSwag | Wino-grande | MMLU | Avg. | Avg. (w/o MMLU) |
|---|---|---|---|---|---|---|---|---|---|
| Mistral-7B (v0.1) | – | 82.1 | 80.9 | 53.8 | 81.0 | 74.0 | 62.4 | 72.4 | 74.4 |
| → LoLCATs (rank=8) | AlpacaClean (+40M) | 81.5 | 81.7 | 54.9 | 80.7 | 74.0 | 51.4 | 70.7 | 74.5 |
| → LoLCATs (rank=8) | RedPajama (+40M) | 80.1 | 77.6 | 49.0 | 80.3 | 71.7 | 53.2 | 68.6 | 71.7 |
| → MVA-SW (rank=32) | AlpacaClean (+20M) | 82.1 | 81.5 | 54.7 | 81.2 | 74.1 | 52.2 | 70.7 | 74.4 |
| → MVA-SW (rank=8) | AlpacaClean (+40M) | **82.3** | **81.9** | **57.6** | 80.2 | 74.0 | 51.6 | 71.2 | **75.2** |
| → MVA-SW (rank=8) | RedPajama (+40M) | 82.5 | 81.5 | 55.7 | 79.7 | 72.9 | 52.4 | 70.8 | 74.5 |

*Table 5.* MVA (MVA-SW without SW) fine-tuning results with inherited Mistral model weights.

**Performance comparison across various 7B models. ♣ denotes models using softmax-attention. † denotes our results.**

| Shot(s) | Size | Tokens | ARC$_e$ 0 | ARC$_c$ 0 | Hella. 0 | PIQA 0 | Wino. 0 | NQ 5 | TriviaQA 5 | MMLU 5 | Avg. |
|---|---|---|---|---|---|---|---|---|---|---|---|
| *Models trained from scratch (for reference)* | | | | | | | | | | | |
| Mistral♣ | 7B | ? | 80.8 | 54.0 | 81.1 | 80.6 | 74.0 | 29.7 | 70.3 | 62.4 | 66.6 |
| *Models finetuned from Mistral 7B* | | | | | | | | | | | |
| SUPRA | 7B | +20B | 74.6 | 42.3 | 74.8 | 80.1 | 67.4 | - | - | 28.0 | - |
| RetNet | 7B | +20B | 73.3 | 39.9 | 72.9 | 77.8 | 66.1 | 16.2 | 43.0 | 26.1 | 51.9 |
| GLA | 7B | +20B | 74.6 | 44.0 | 75.9 | 79.2 | 69.5 | 22.2 | 57.8 | 28.4 | 56.5 |
| GSA | 7B | +20B | 75.9 | 43.9 | 76.5 | 78.7 | 70.1 | 23.4 | 60.7 | 32.4 | 57.7 |
| MVA(Ours)† | 7B | +2.5B | 75.7 | 44.9 | 76.0 | 80.4 | 69.7 | 21.2 | 59.1 | 30.2 | 57.2 |
| MVA(Ours)† | 7B | +10B | **78.3** | **47.5** | **78.1** | **80.5** | **72.1** | **25.9** | **65.9** | **34.4** | **60.3** |

200 steps of linear warm-up, and a training length of 2K steps with a batch size of 0.5M tokens. The dataset used is the SlimPajama (Soboleva et al., 2023) corpus. The results are shown in Table 5.

Our MVA method achieves performance comparable to state-of-the-art GSA after fine-tuning with 2.5B tokens. After fine-tuning with 10B tokens, MVA surpasses GSA in all tasks. This demonstrates that our MVA introduces smaller errors compared to GSA and GLA, enabling a rapid recovery of the original Attention's performance. The effectiveness of this observation is further supported by ablation studies.

## 4.4. MVA Inference Efficiency

Table 6 demonstrates the inference efficiency of MVA. In terms of memory consumption, MVA performs comparably to GSA. However, due to its vocabulary decomposition requirement, MVA exhibits slightly higher generation latency than GSA. Notably, both MVA and GSA significantly outperform FlashAttention in all efficiency metrics and memory usage when the sequence length reaches 64K.

## 4.5. Ablation Studies

Finally, we examine the impact of incorporating the sliding window. The results show that adding a sliding window significantly preserves the performance of the original At-

*Table 6.* Inference efficiency comparison of MVA, GSA, and FlashAttention under different sequence lengths. OOM indicates out-of-memory errors.

| Model | Seq Len | Full Inf Time (s) | Full Inf Mem (GB) | Prefill Time (s) | Gen Latency (ms/token) | Total Mem (GB) |
|---|---|---|---|---|---|---|
| MVA | 4K | 0.14 | 0.77 | 0.125 | 98.8 | 15.32 |
| | 8K | 0.26 | 1.53 | 0.249 | 60.3 | 16.58 |
| | 16K | 0.51 | 3.06 | 0.508 | 78.8 | 19.08 |
| | 32K | 1.08 | 6.11 | 1.090 | 79.8 | 24.09 |
| | 64K | 2.25 | 12.22 | 2.265 | 97.3 | 34.11 |
| | 128K | 5.08 | 24.44 | 7.156 | 58.1 | 54.14 |
| GSA | 4K | 0.10 | 0.76 | 0.077 | 93.2 | 15.30 |
| | 8K | 0.17 | 1.53 | 0.153 | 40.9 | 16.55 |
| | 16K | 0.32 | 3.06 | 0.315 | 48.5 | 19.06 |
| | 32K | 0.64 | 6.11 | 0.630 | 63.0 | 24.07 |
| | 64K | 1.29 | 12.22 | 1.293 | 90.2 | 34.08 |
| | 128K | 2.66 | 24.44 | 5.102 | 38.7 | 54.11 |
| Flash Attention | 4K | 0.05 | 1.26 | 0.056 | 21.7 | 15.79 |
| | 8K | 0.11 | 2.53 | 0.116 | 27.5 | 18.54 |
| | 16K | 0.27 | 5.05 | 0.287 | 46.3 | 23.55 |
| | 32K | 0.73 | 10.11 | 0.750 | 92.4 | 33.55 |
| | 64K | 2.19 | 20.22 | 2.208 | 220.4 | 53.57 |
| | 128K | | | *OOM* | | |

tention, while reducing the training resources. However, the standalone sliding window is limited by its window size. Integrating MVA enhances its performance, achieving at least a 5-point improvement on MMLU tasks.

We investigate the contributions of hierarchical integration and multi-level vocabulary decomposition. First, replacing the Softmax function in GSA with a perceptron significantly improves performance. Integrating MetaLA further enhances performance, with our hierarchical integration out-

*Table 7.* Impact of sliding window on MVA-SW performance.

| Component | MMLU |
|---|---|
| MVA-SW (soft bond) | 57.1 |
| MVA-SW (hard bond) | 53.2 |
| Sliding Window Only | 51.9 |
| noSW | 25.0 |

performing both GSA and MetaLA, achieving at least a 3.3-point improvement on ARC-Challenge and a 0.9-point improvement on ARC-Easy. Finally, replacing the memory strategy with multi-level vocabulary decomposition further boosts performance, yielding a 1.9-point gain on ARC-Challenge and a 3.2-point gain on ARC-Easy. The results

*Table 8.* Ablation study results on ARC dataset. The table compares different methods and their performance on ARC, as well as the token budget (in billions). VD means vocabulary decomposition.

| Method | arc_challenge | arc_easy | Tokens (B) |
|---|---|---|---|
| MVA w/ 3 order (GSA) | 0.3389 | 0.6596 | 0.8 |
| MVA w/ 3 order (GSA) | 0.3763 | 0.6949 | 2.0 |
| MVA w/ 3 order (GSA-sigmoid) | 0.3407 | 0.6732 | 0.8 |
| MVA w/ 1 order (MetaLA) | 0.3527 | 0.6987 | 0.8 |
| MVA w/o VD (GSA+MetaLA) | 0.3857 | 0.7075 | 0.8 |
| MVA w/ VD (attn(K,mk,mk)) | 0.3906 | 0.7210 | 0.8 |
| MVA w/ VD (delta rule-like) | 0.4044 | 0.7391 | 0.8 |

indicate that our method effectively combines the strengths of GSA and MetaLA, achieving substantial performance improvements. Multi-level vocabulary decomposition further reduces the error relative to Softmax Attention.

*Table 9.* Performance comparison of different models. Attention replacing Mistral-7B with a training length of 2K and a gradient accumulation of 128. VD means vocabulary decomposition and we use a 2-level word list decomposition.

| Method | Memory Usage (GPU) | Time per Iteration |
|---|---|---|
| MetaLA | 36317MiB | 75.08 s/it |
| GSA | 37619 MiB | 81.67 s/it |
| MVA w/o VD | 38885 MiB | 82.18 s/it |
| MVA w/ VD | 40096 MiB | 105.79 s/it |

## 5. Conclusion

We propose a unified linear attention framework that integrates MetaLA, GSA, and Delta Rule-inspired methods using high-order $QK$ integration and multi-level vocabulary decomposition theories.

The high-order $QK$ integration theory reveals that GSA focuses on high-frequency information through higher-order $QK$-terms, while MetaLA captures low-frequency information using lower-order $QK$-terms. These complementary

behaviors allow our method to exploit both perspectives for enhanced performance. In addition, multi-level vocabulary decomposition recursively reduces information loss, achieving exponential memory capacity expansion and improving approximation accuracy for high-rank matrices.

MVA outperforms GSA by 2.3 points with only half the training resources and consistently excels across all test sets. Unlike LoLCATs two steps, MVA-SW simplifies fine-tuning and achieves superior MMLU performance with fewer training tokens, demonstrating its efficiency and effectiveness.

## Impact Statement

This paper presents work whose goal is to transform pretrained Softmax-based language models into linear attention models. This transformation technique utilises lower costs to train more efficient language models. Therefore this technique might help democratize access of language models. Whether the efficient transformation would affect known problems, such as biased and harmful outputs of language models remains an unexplored research question.

## Acknowledgements

This research is supported by the National Science and Technology Major Project of China (2022ZD0116316). This work was partially supported by National Natural Science Foundation of China (62325603, 62236009, U22A20103), CAS Project for Young Scientists in Basic Research (YSBR-116), Beijing Science and Technology Plan (Z241100004224011). We are also thankful to all the reviewers for their insightful comments and rigorous evaluation.

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

*Table 10.* Performance comparison on long-context benchmarks (Qasper, NarrativeQA, QMSum). Top section shows models trained from scratch; bottom section shows methods fine-tuned from Mistral-7B.

| Model | Qasper | NarrativeQA | QMSum |
|---|---|---|---|
| **Models Trained from Scratch** | | | |
| RWKV6 | 9.2 | 14.4 | 1.1 |
| Mamba | 5.6 | 27.9 | 0.8 |
| Mistral | 25.8 | 25.1 | 5.0 |
| **Fine-tuned from Mistral-7B (20B tokens)** | | | |
| RetNet | 11.1 | 0.0 | 0.0 |
| GLA | 18.4 | 17.2 | 9.0 |
| GSA | 18.8 | 19.2 | 10.0 |
| MVA | **20.7** | **20.4** | 9.58 |

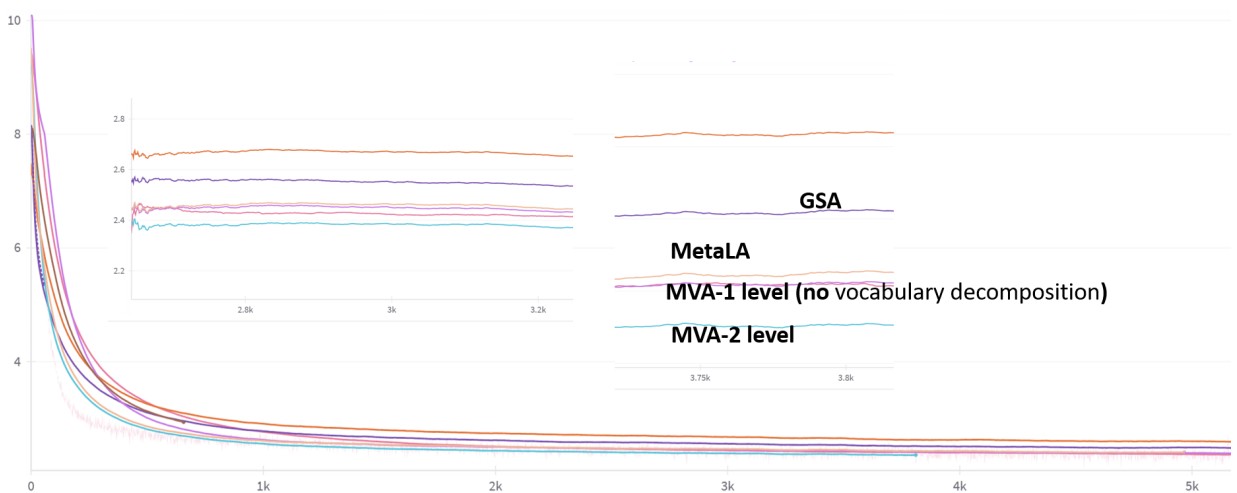

*Figure 2.* Speed of convergence for different models.

## A. Fine-tuning the loss curve

In this subsection, we show the convergence speed of MVA, MVA (wordless table decomposition), GSA, and MetaLA, as shown in Figure 2, it is obvious that the convergence speed of MVA is very fast and much better than GSA.In addition, we show the loss of the whole fine-tuning process of MVA, and our method is very stable in the fine-tuning process and the convergence tendency is very obvious, as shown in Figure 3.

### A.1. MVA performance on long sequence tasks

We compared them on the long sequence tasks listed by GSA, as shown in Table 10.

## B. Related Work

### B.1. Linear Attention

Full attention in Transformers (Vaswani et al., 2017) and other efficient Attention (Zaheer et al., 2020; Roy et al., 2020; Mohtashami & Jaggi, 2023; Zhu et al., 2023; Kitaev et al., 2020; Child et al., 2019; Yun et al., 2021; Beltagy et al., 2020) that can't be recursive requires a KV cache that scales linearly with sequence length, resulting in significant computational and memory overhead during inference for long sequences. This limitation often makes it infeasible to process extremely long sequences. Linear attention (LA) (Katharopoulos et al., 2020; Shen et al., 2021) addresses this issue by replacing the softmax operation in attention with a kernel function and reordering operations. Specifically, it first computes $\phi(K)V$ to obtain a fixed-size state $S$, and then computes $\phi(Q)S$ to produce the output. This method achieves linear complexity and

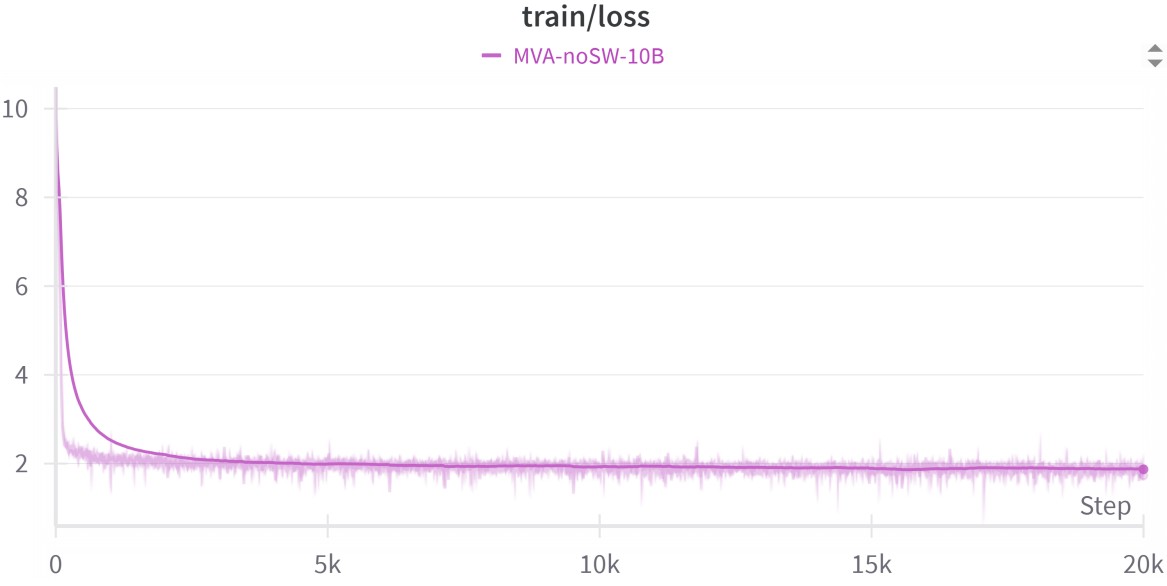

*Figure 3.* Fine-tuned loss curves for MVA based on Mistral-7B modeling.

enables inference with a fixed-size state, regardless of sequence length, by maintaining the state recursively.

However, linear attention has several issues, including insufficient focus on relevant tokens, the inability to erase outdated information during state updates, and a lack of bias toward closer tokens. To address these problems, Gated Linear Attention (GLA) reintroduced gating mechanisms. During state updates, the gate selectively erases irrelevant information before incorporating new data, which also introduces a dynamic decay mechanism to emphasize local information. GLA parameterizes many linear Attention, SSM models, and other well-designed recursive models into their GLA form for uniform system-level optimization. Building on GLA, MetaLA analyzed existing linear models, classifying current linear models into three categories: the LinFormer (Sun et al., 2023; Wang et al., 2020b; Qin et al., 2022), SSM (Gu & Dao, 2024; Gupta et al., 2022; Smith et al., 2023; Fu et al., 2023), and LinRNN (De et al., 2024; Hanada et al., 2023) and proposed four necessary conditions for optimal linear approximations to softmax attention. Based on these conditions, MetaLA achieves state-of-the-art performance among linear models.

Other approaches, such as Flatten Transformer (Han et al., 2023) and Performer (Peng et al., 2021), explore alternative kernel functions to replace softmax for more focused attention. Additionally, non-attention architectures like SSM-based models (e.g., S4, H3 (Fu et al., 2023)), RetNet (Sun et al., 2023), RWKV-4 (Peng et al., 2023), and HGRN (Qin et al., 2023), as well as LRU (Yue et al., 2023), can be viewed as special cases of GLA or MetaLA. In some scenarios, these models converge to MetaLA. Despite these advancements, linear attention models often suffer from a lack of high-frequency information, corresponding to the higher-order terms in the Taylor expansion of softmax. This limitation creates a significant gap with softmax attention, hindering their compatibility with existing LLM weights for fine-tuning. Consequently, recent research focuses on developing linear models that retain softmax while enabling recursive mechanisms.

### B.2. Sparse Attention

Sparse attention methods, such as ABC and Gated Slot Attention (GSA) (Zhang et al., 2024b), retain the softmax function, preserving most of the frequency information in Q and K and achieving closer approximations to full softmax attention. ABC (Peng et al., 2022) introduces learnable pseudo-queries for compressing the KV sequences, storing fixed-size states. However, it lacks mechanisms to erase irrelevant information, leading to accumulation of unnecessary data. GSA addresses this by incorporating a gating mechanism in the KV state updates, enabling controlled writes to the state and more efficient compression.

Although sparse attention methods approximate softmax more effectively, they lack sufficient low-frequency information,

introducing non-negligible errors. Combining linear and sparse attention approaches can further reduce the gap with softmax attention. However, both approaches face challenges in efficiently expanding memory capacity. Inspired by the Delta Rule, our work introduces a mechanism to dynamically expand memory capacity.

### B.3. Delta Rule Methods

Fast Weight Programmers (Schlag et al., 2021) proposed a more precise update mechanism that first queries the previous state using the current $K$, removes the queried information from the current input, and updates only with new information. This maximizes utilization of the limited state space. Parallels Delta Rule (Yang et al., 2025; 2024a) further parallelized this mechanism, significantly improving training speed.

### B.4. Transformer-to-RNN Conversion

Linear models offer substantial advantages over full attention, including a fixed-size state space and faster inference for longer tasks. However, training linear models from scratch demands significant resources. Transformer-to-RNN (T2R) (Kasai et al., 2021) conversion addresses this by fine-tuning existing LLM weights with softmax attention to transform them into linear models. Notable methods, such as SUPRA (Mercat et al., 2024) and GSA, have demonstrated promising results. Recently, LoLCATs (Zhang et al., 2024a) proposed a two-step process involving attention transfer and low-rank linearization (Hu et al., 2021), combined with a sliding window mechanism. This approach transforms LLMs into linear models with nearly linear complexity while preserving most of their performance.

## C. Error Analysis with Softmax Attention

This section analyzes the error between our method and Softmax Attention, demonstrating that our approach achieves superior approximation in larger memory capacities compared to other methods. From Section 3.3, we know that the MVA method can fully converge to GSA and MetaLA. When they are optimal, the parameters $W_{qc}^{(i)}$, $W_S^{(i)}$, and $W_{kc}^{(i)}$ converge to very small values, such that their impact on the output is negligible. Consequently, the solution space of our method almost entirely encompasses the above two methods, and their unified combination enables convergence to better solutions. This unification ensures complementary convergence rather than mutual interference. The following provides a detailed error analysis:

Original Attention:

$$O = \text{Softmax}\left((Q \odot P)(K \odot P)^\top \odot M\right) V$$

Our MVA computation is given by:

$$Q_{st} = \text{Softmax}\left(\left((((Q \odot B)\left(\frac{K}{B}\right)^\top) \odot M)(1 - G)\right) W_S + \phi(Q)\right)$$

$$O = (((Q_{st} \odot B)\left(\frac{1 - G}{B}\right)^\top) \odot M)V$$

### C.1. Influence of High-order Integration

The error between our method and Softmax Attention can be expressed as follows:

$$\text{Softmax}\left((Q \odot P)(K \odot P)^\top \odot M\right) V - (((Q_{st} \odot B)\left(\frac{1 - G}{B}\right)^\top) \odot M)V$$

Focusing on the error between specific attention scores:

The expression $e^{q_i k_j^\top}$ can be expanded as:

$$e^{q_i k_j^\top} = \sum_{m=0}^{\infty} \frac{1}{m!} \sum_{s_1+s_2+\cdots+s_d=m} \binom{m}{s_1, s_2, \ldots, s_d} \prod_{l=1}^{d} (q_{il})^{s_l} (k_{jl})^{s_l} \tag{13}$$

$$= \sum_{m=0}^{\infty} \sum_{s_1+s_2+\cdots+s_d=m} \prod_{l=1}^{d} \frac{(q_{il} k_{jl})^{s_l}}{s_l!} \tag{14}$$

$$= \sum_{m=0}^{\infty} \sum_{s_1+s_2+\cdots+s_d=m} F_{s_l}(q_{il}) \prod_{l=1}^{d} (k_{jl})^{s_l} \tag{15}$$

$$= \sum_{s_1,s_2,\ldots,s_d} F_{s_l}(q_{il}) \prod_{l=1}^{d} (k_{jl})^{s_l} , \tag{16}$$

where $F_{s_l}(q_{il}) = \frac{(q_{il})^{s_l}}{s_l!}$.

The MetaLA can be further expressed as:

$$\phi(q_i) e^{-(-Ck_j^\top)} = \sum_{t=0}^{\infty} \frac{1}{t!} \phi(q_i)(Ck_j^\top)^t \tag{17}$$

$$= \sum_{t} \phi(q_{it}) \left( \sum_{s_1,s_2,\ldots,s_d} F_{sl}(C_{tl}) \prod_{l=1}^{d} (k_{jl})^{s} \right) \tag{18}$$

$$= \sum_{s_1,s_2,\ldots,s_d} \left( \sum_{t} \phi(q_{it}) F_{sl}(C_{tl}) \right) \prod_{l=1}^{d} (k_{jl})^{s} , \tag{19}$$

where $\phi(x) = x$.

The error between Full Attention and MetaLA is given by:

$$e^{q_i k_j^\top} - \phi(q_i) e^{-(-Ck_j^\top)} = \sum_{s_1,s_2,\ldots,s_d} \left( F_{sl}(q_{il}) - \sum_{t} \phi(q_{it}) F_{sl}(C_{tl}) \right) \prod_{l=1}^{d} (k_{jl})^{s} . \tag{20}$$

This can be interpreted as using a fixed function to fit different powers of $F_{sl}(q_{il})$. Since MetaLA adopts $\phi(x) = x$, it can only fit first-order information. To further reduce the error, $Q$ needs to be extended to higher powers. However, due to the fixed power of the $\phi$ function, it can only eliminate fixed patterns of frequencies.

To minimize the error further, higher-order $q$ terms need to be introduced for different powers of $k$. One method involves increasing the order of $k$ and querying with at least the same order of $Q$:

$$\sum_{i=1}^{m} \phi_i(Q) F_{i1}(K)^\top F_{i1}(K) \ldots F_{ii}(K)^\top, \tag{21}$$

where different $i$ terms do not affect the lower-order terms, ensuring that this approach can achieve at least $m$-order fitting, leaving only $m+1$-order errors.

Another method is to avoid explicit summation by considering introducing dynamically varying powers of $Q$ based on the powers of $K$. This requires embedding $Q$ and $K$ within a higher-order nonlinear function, such as $e^{QK^\top F(X)}$. Here, $F(X)$ compresses $K^\top$ to a fixed size, enabling linear complexity and recursive inference. This type of high-order nonlinear function enhances the previous method by introducing more powers for further approximating the original Attention but inevitably introduces significant high-frequency noise. GSA adopts this strategy, the GSA attention mechanism, given by:

$$\text{Softmax}(Q(F(CK_n^\top)K)^\top)F(CK_n^\top)V,$$

has its attention score at position $(i,j)$ expressed as:

$$e^{(q_i \sum_{a}^{i}(F(Ck_a^\top)k_a)^\top)}F(Ck_j^\top).$$

Expanding this, we obtain:

$$\left( \sum_{s_1,s_2,\ldots,s_d} \sum_t \left( \exp\left( \sum_{a=0}^{i} q_i k_a^\top \mathrm{F}_{it}(k_a C^\top) \right) \mathrm{F}_{sl}(C_{tl}) \right) \prod_{l=1}^{d} k_{jl}^s \right) \tag{22}$$

$$= \left( \sum_{s_1,s_2,\ldots,s_d} \sum_t \left( \left( \sum_u \frac{1}{u!} \left( q_i \sum_{a=0}^{i} k_a^\top \mathrm{F}_{it}(k_a C^\top) \right) \right)^u \mathrm{F}_{sl}(C_{tl}) \right) \prod_{l=1}^{d} k_{jl}^s \right), \tag{23}$$

where $\mathrm{F}(C k_a^\top)$ is the transformation applied to the context. By approximating $\sum_{a=0}^{i} k_a^\top \mathrm{F}_t(k_a C^\top)$ as an averaged matrix $\bar{C}_a$, where $\bar{C}_a$ retains only higher-order $k$ terms, the expression becomes:

$$\left( \sum_{s_1,s_2,\ldots,s_d} \sum_t \left( \sum_u \frac{1}{u!} (q_i \bar{C}_{at})^u \mathrm{F}_{sl}(C_{tl}) \right) \prod_{l=1}^{d} k_{jl}^s \right) \tag{24}$$

$$= \left( \sum_{s_1,s_2,\ldots,s_d} \sum_t \left( \sum_{u_1,u_2,\ldots,u_d} \mathrm{F}_{ul}(q_i) \prod_{l=1}^{d} (\bar{C}_{at})^u \mathrm{F}_{sl}(C_{tl}) \right) \prod_{l=1}^{d} k_{jl}^s \right). \tag{25}$$

Focusing on the $\prod_{l=1}^{d} k_{jl}^s$ basis, the terms involving $q$ and $\bar{C}_a$ have the same order. To mitigate the influence of high-frequency noise introduced by $\prod_{l=1}^{d} k_{jl}^s$, GSA tends to prioritize high-frequency information. Thus, we consider incorporating methods such as MetaLA or GLA to balance and supplement the information.

The discrepancy between GSA and Softmax Attention is given by:

$$e^{q_i k_j^\top} - e^{(q_i \sum_a^i (\mathrm{F}(C k_a^\top) k_a)^\top)} \mathrm{F}(C k_j^\top) \tag{26}$$

$$= \sum_{s_1,s_2,\ldots,s_d} \left( \mathrm{F}_{sl}(q_i) - \sum_t \left( \sum_{u_1,u_2,\ldots,u_d} \mathrm{F}_{ul}(q_i) \prod_{l=1}^{d} (\bar{C}_{at})^u \right) \mathrm{F}_{sl}(C_{tl}) \right) \prod_{l=1}^{d} k_{jl}^s. \tag{27}$$

This represents a dynamic average approximation of the full attention $F_{sl}(q_i)$ over $q$'s various orders. Expanding GSA further:

$$\sum_{s_1,s_2,\ldots,s_d} \left( \sum_{u_1,u_2,\ldots,u_d} \sum_v \mathrm{F}_{vul}(q_i) \bar{\mathrm{F}}(C) \right) \mathrm{F}_{sl}(C_{tl}) \prod_{l=1}^{d} k_{jl}^{s+u} \tag{28}$$

$$= \sum_{s_1,s_2,\ldots,s_d} (\bar{\mathrm{F}}_{vul}(q_i) \bar{\mathrm{F}}(C) \mathrm{F}_{sl}(C_{tl})) \prod_{l=1}^{d} k_{jl}^s. \tag{29}$$

Introducing methods such as MetaLA transforms the equation into our MVA-no vocabulary decomposition, which can be expressed as:

$$\sum_{s_1,s_2,\ldots,s_d} (\phi(q_i) + \bar{\mathrm{F}}_{vul}(q_i) \bar{\mathrm{F}}(C) \mathrm{F}_{sl}(C_{tl})) \prod_{l=1}^{d} k_{jl}^s. \tag{30}$$

The additional term $\phi(q_i)$ alleviates part of GSA's fitting pressure, suppressing high-frequency noise:

$$\mathrm{F}_{sl}(q_i) = (\mathrm{F}_{sl}(q_i) - \phi(q_i)) (\bar{\mathrm{F}}(C) \mathrm{F}_{sl}(C_{tl}))^{-1}.$$

Compared to the original formulation:

$$\mathrm{F}_{sl}(q_i) = \mathrm{F}_{sl}(q_i) (\bar{\mathrm{F}}(C) \mathrm{F}_{sl}(C_{tl}))^{-1},$$

higher-order terms are similarly suppressed as $\bar{F}(C) F_{sl}(C_{tl})$ decreases, mitigating high-frequency noise.

C.1.1. NOT USING THE TAYLOR SERIES EXPANSION

Consider not using the Taylor series expansion, directly with full Attention for differential exhaustion error analysis, the process is as follows: - Original Attention:

$$e^{q_i k_j^\top} = 1 + \sum_m \frac{1}{m!}(q_i k_j^\top)^m$$
$$= 1 + \sum_{m=odd} F_m(q_i)k_j^\top(k_j k_j^\top)^{m-1} + \sum_{s=even} F_s(q_i)(k_j k_j^\top)^s + F(k_j).$$

- GSA Attention Scores:

$$s_{ij} = \sum_s e^{q_i \hat{k}_s^\top} \prod_{t=i-j+1}^i \alpha_t(1-\alpha_{sj})^\top = \sum_s e^{q_i(\prod_{t=i-j+1}^i \alpha_t k_s^\top(1-\alpha_s))+log(\prod_{t=i-j+1}^i \alpha_t(1-\alpha_{sj})^\top)} = e^{q_i f(k_1,...,k_j)+f_{g1}(k_1,...,k_j)}$$

$$e^{a_1 x_1 + a_2 x_2 + \cdots + a_n x_n} = \frac{e^{x_1} + e^{x_2} + \cdots + e^{x_n}}{n}.[a_1, a_2, ..., a_n] = ln(\sum_i e^{x_i}/n)X^\dagger$$

where $X^\dagger = (X^\top X)^{-1}X^\top$ is the pseudo-inverse of $X$.

- MetaLA Attention Scores:

$$m_{ij} = \frac{\phi(q_i)\prod_{t=i-j+1}^i \alpha_t e^{-k_j^\top W}}{1+e^{-k_j^\top W}} = \phi(q_i)e^{f_{m2}(k_j,...,k_i)}$$

Rewriting the GSA scores as:

$$s_{ij} = e^{q_i f(k_1,...,k_j)} = e^{q_i \bar{k}_j^\top + f_{g1}(k_1,...,k_j)},$$

where $\bar{k}_j$ represents a weighted approximation of the original $k_j$ due to limited memory capacity. The error between GSA and the original attention is:

$$\epsilon_{ij} = e^{q_i k_j^\top} - e^{q_i \bar{k}_j^\top} = e^{q_i \bar{k}_j^\top}\left(e^{q_i(k_j^\top - \bar{k}_j^\top)} - e^{f_{g1}(k_1,...,k_j)}\right).$$

Using a Taylor expansion of $e^{q_i(k_j^\top - \bar{k}_j^\top)}$ around the first-order term:

$$1 + \sum_m (q_i(k_j^\top - \bar{k}_j^\top))^m,$$

because $e^{f_1(k_1,...,k_j)} \approx 1$, we find that the error can be approximated as:

$$\epsilon_{ij} \approx e^{q_i \bar{k}_j^\top}q_i(k_j^\top - \bar{k}_j^\top).$$

Introducing MetaLA further reduces the error, as:

$$\epsilon_{ij} \approx q_i\left(e^{q_i \bar{k}_j^\top + log(k_j^\top - \bar{k}_j^\top)} - e^{f_{m1}(q_i)+f_{m2}(k_j,...,k_i)}\right).$$

This result indicates that the use of $f_1(q_i) + f_2(k_j, \ldots, k_i)$ can effectively minimize the residual error. A similar analysis holds when including the denominator.

Additionally, starting from MetaLA, the error can be expressed as:

$$1 + \sum_{m=odd} F_m(q_i)k_j^\top(k_j k_j^\top)^{m-1} + \sum_{s=even} F_s(q_i)(k_j k_j^\top)^s + F(k_j) - \phi(q_i)f_3(k_j)\prod_{t=i-j+1}^i \alpha_t.$$

$$1 + \sum_{m=odd} F_m(q_i)k_j^\top(k_j k_j^\top)^{m-1} + \sum_{s=even} F_s(q_i)(k_j k_j^\top)^s + F(k_j) - F(\phi(q_i)f_{g1}(k_j)^\top f_{g2}(k_j)\prod_{t=i-j+1}^i \alpha_t)f_{g1}(k_j)^\top\prod_{t=i-j+1}^i \alpha_t.$$

MetaLA first removes the first-order terms in the attention map related to $q$, which is crucial because first-order terms contribute significantly to the overall attention mechanism's performance. Subsequently, GSA's dynamic weighted approximation mitigates higher-order terms, limited by its memory constraints.

C.1.2. DYNAMIC DECAY

For gated attention models, we use the following formulation:

$$G_n^\top = F_a(CK_n^\top), \quad B_n^\top = F_b(G_n^\top),$$

where $K_n \in \mathbb{R}^{n \times d_k}$ is key tokens and $C \in \mathbb{R}^{m \times d_k}$ is a learnable parameter representing $m$ pseudo-query tokens to query $K_n$, with the results applied to $K_n$ or $V_n$ sequences. Here, $m$ effectively defines the memory capacity. $G_n$ is gating matrix and $B_n$ is decay matrix in 1, 2 and 3. For models like MetaLA, GLA, and GSA, $G_n^\top$ is typically sigmoid$(CK_n^\top)$, and $B_n^\top$ is $\exp(\log(G_n^\top)U_n^\top)$, where $U_n$ is an lower triangular matrix of shape $n \times n$. For ABC, $F_a$ is Softmax, and $B_n^\top$ is $\exp(\log(G_n^\top)O_n^\top)$, where $O$ is a zero matrix of shape $n \times n$.

For MetaLA, the memory mechanism is given by:

$$[B_n \cdot (1 - A_n)]V_n,$$

where $1 - A_n$ represents attention integration over $V$ using first-order pseudo-query attention, and $B_n$ adjusts this attention. Based on these characteristics, we define two attributes for MetaLA: first-order learnable pseudo-query attention over $K$ and first-order adjustment-based memory, referred to as a **complete first-order $K$ integration method**. Higher-order terms arise from powers of $K$, but the information primarily originates from first-order $K$.

For GSA, the mechanism involves $[B_n \cdot (1 - A_n)]K_n$ and further processes $[B_n \cdot (1 - A_n)]V_n$ to compute the final state. This corresponds to integrating $K$ and $V$ with first-order pseudo-query attention, then re-integrating $V$ using the resulting $K$. Since this involves third-order terms of $K$, it is referred to as a **complete third-order $K$ integration method**, albeit without branches for first-order and second-order $K$ integration. Thus, we introduce MetaLA to complement first-order integration.

Causal convolutions in these methods can be interpreted as zero-order integration. Consequently, we propose a full-branch high-order $K$ integration method, detailed in the Appendix. In summary, high-order methods effectively approximate the higher-order terms of the Taylor expansion for Softmax attention.

## C.2. Influence of Multi-level Vocabulary Decomposition

**Theorem C.1** (Improved Memory Capacity). *After $c$ recursions, the expected error $E$ can be bounded as follows:*

$$\|E\| \leq \|K\| \prod_{i=0}^{c} \epsilon_i \prod_{i=0}^{c} \gamma_i,$$

*where $\epsilon_i = \frac{n-m}{n}$ and $\gamma_i$ is determined by learnable parameters that ensure convergence to a value less than 1.*

**proof:** For the GSA-like component in MVA, as shown in Equation (11), if we remove the $\phi(Q)$ branch and treat the matrix $B$ as learnable relative positional encoding, the GSA branch output is:

$$O = \left(\text{Softmax}\left((QK^\top \odot M)F(K)\right)W_S F(K)^\top \odot M\right)V,$$

where $F(K) \in \mathbb{R}^{n \times m}$, with $m$ as the memory size and $n$ as the sequence length. Considering extreme cases such as quantization functions or Softmax, $F(K)$'s row vectors approach a one-hot distribution, effectively selecting specific elements from the attention map.

Extracting $F(K)$ equivalently from the Softmax yields:

$$\text{Softmax}((QK^\top \odot M)F(K)) = \text{Softmax}((QK^\top \odot M))\hat{F}(Q, K),$$

where:

$$\hat{F}(Q, K) = [\text{Softmax}((QK^\top \odot M))]^{-1}\text{Softmax}((QK^\top \odot M)F(K)),$$

and $\hat{F}(Q, K)$ has dimensions $n \times m$.

Thus, the output becomes:

$$O = \left(\text{Softmax}((QK^\top \odot M))\hat{F}(Q, K)W_S F(K)^\top \odot M\right)V.$$

Compared to the original Softmax Attention output, this introduces a dynamic low-rank matrix $\hat{\mathrm{F}}(Q, K)W_S\mathrm{F}(K)^\top$. Ideally, if this matrix is equivalent to an identity matrix, the error is zero. However, achieving this is challenging due to rank limitations, and maximizing the matrix rank is essential to minimize the error.

We first consider the error term in the following form:

$$\epsilon = \Big(\text{Softmax}((QK^\top \odot M))(I - \hat{\mathrm{F}}(Q, K)W_S\mathrm{F}(K)^\top) \odot M\Big) V.$$

Next, we decompose this error as:

$$\epsilon = \Big(\text{Softmax}((QK^\top \odot M))(I - \hat{\mathrm{F}}(Q, K)W_S \sum_i \mathrm{F}^{(i)}(K)^\top) \odot M\Big) V^{(i)}.$$

We focus on reducing the above error term and approach this problem from two key aspects:

**1. Reducing the Error from the Term** $(I - \hat{\mathrm{F}}(Q, K)W_S\mathrm{F}(K)^\top)$**:**

The term $(I - \hat{\mathrm{F}}(Q, K)W_S\mathrm{F}(K)^\top)$ involves the matrices:

$$\hat{\mathrm{F}}(Q, K) \in \mathbb{R}^{n \times m}, \quad W_S \in \mathbb{R}^{m \times m}, \quad \mathrm{F}(K)^\top \in \mathbb{R}^{m \times n}.$$

For large sequence lengths, the product $\hat{\mathrm{F}}(Q, K)W_S\mathrm{F}(K)^\top \in \mathbb{R}^{n \times n}$ results in a low-rank matrix of rank $m$. To minimize the error in this term, we aim to make this low-rank matrix close to the identity matrix, both in terms of rank and numerical values.

**Choice of the F-function**

To ensure that the low-rank matrix approximates the identity matrix, we require the row vectors of the matrices $\hat{\mathrm{F}}(Q, K)W_S'$ and $\mathrm{F}(K)W_S' \in \mathbb{R}^{n \times m}$ to be aligned in the same positions, with orthogonality between different positions. The simplest form of orthogonal matrices is the identity vector. Although this might seem like a special case, it is a generalizable approach. After learning other orthogonal vectors for the matrices $\hat{\mathrm{F}}$ and $\mathrm{F}$, the matrix $W_S'$ can transform them into identity vectors. Moreover, the current methods using $\hat{\mathrm{F}}$ and $\mathrm{F}$ exhibit a tendency to approach identity vectors.

Thus, the condition for numerical computation to approach the identity matrix suggests that the F-function should be extremal, such as the delta function or exponential functions. These functions make learning and convergence easier. Subsequently, normalization via a parameter matrix or other methods can be applied, which results in extreme scaling of the matrix values, where dominant values approach 1, and other values approach 0. This results in a matrix $\hat{\mathrm{F}}(Q, K)W_S', \mathrm{F}(K)W_S' \in \mathbb{R}^{n \times m}$ that resembles the following form:

$$\begin{bmatrix} sv_{11} & sv_{12} & \dots & \dots & maxv_1 & \dots & sv_{1n} \\ sv_{21} & sv_{22} & \dots & maxv_2 & \dots & \dots & sv_{2n} \\ \vdots & \vdots & \ddots & \vdots & \vdots & \vdots & \vdots \\ sv_{m1} & maxv_m & \dots & \dots & \dots & \dots & sv_{mn} \end{bmatrix}^T \rightarrow_{\mathrm{F}} \begin{bmatrix} 0 & 0 & \dots & \dots & 1 & \dots & 0 \\ 0 & 0 & \dots & 1 & \dots & \dots & 0 \\ \vdots & \vdots & \ddots & \vdots & \vdots & \vdots & \vdots \\ 0 & 1 & \dots & \dots & \dots & \dots & 0 \end{bmatrix}^T$$

For the matrices $A = \hat{\mathrm{F}}(Q, K)W_S'$ and $B = \mathrm{F}(K)W_S'$, the maximum values should appear at the same positions to make the matrix approach the identity matrix $I$. This implies that corresponding rows must be similar, while different rows should ideally be orthogonal. For the matrix product $AB^\top$, the value at position $(i, j)$ is given by:

$$(AB^\top)_{ij} = \sum_t a_{it}b_{jt}.$$

For the matrices $A$ and $B$ to approach the identity matrix, we need:

$$\sum_t a_{it} b_{it} = 1.$$

However, this introduces additional error when the positions of the maximum values in different rows of $A$ and $B$ coincide, leading to $\sum_t a_{it} b_{it} = 1$, while in the identity matrix, $\sum_t I_{ij} = 0$.

**Ensuring Consistency in the Maximum Value Positions**

To ensure that the maximum values occur at the same positions in both matrices $A$ and $B$, we leverage the structure of the $\mathrm{F}(CK^\top)$ function used in MVA. Specifically, when the vector in $K$ is most related to the "query token" in $C$, it will yield larger values, which are then amplified by the exponential function in $F$, causing the corresponding position to take on the maximum value.

In our implementation, we use sigmoid and softmax functions with exponential methods. Additionally, more extreme approaches could involve directly assigning a value of 1 to the most relevant position and 0 to other positions, as described in Section 3.2 with the $Q$-function.

For the matrix $A = \hat{\mathrm{F}}(Q, K)W_S'$, which also depends on $Q$, compared to the function $\mathrm{F}(CK^\top)$ that is independent of $Q$, the dynamic nature of $A = \hat{\mathrm{F}}(Q, K)W_S'$ can lead to larger errors in certain scenarios. Therefore, future research will focus on incorporating recursive updates in $\mathrm{F}(CK^\top)$, allowing it to dynamically adjust based on $Q$ to generate orthogonal vectors at the same positions as those in $A = \hat{\mathrm{F}}(Q, K)W_S'$.

**2. Exponential Power Increase of Memory Capacity for Rank Augmentation**

For low-rank matrices, let us define:

$$L = \hat{\mathrm{F}}(Q, K)W_S'{W_S'}^\top \mathrm{F}(K)^\top = \mathrm{Softmax}(QK^\top)^{-1}\mathrm{Softmax}(QK^\top(1-\mathrm{Sigmoid}(KC^\top)))W_S'{W_S'}^\top(1-\mathrm{Sigmoid}(KC^\top))^\top$$

The rank of the above matrix is determined by the parameter matrix $C \in \mathbb{R}^{d \times m}$. A simple way to increase the rank is to manually set the value of $m$ in the matrix $C$, for example, $C \in \mathbb{R}^{d \times (2m)}$. This method increases both the memory capacity and the rank to $2m$, but it is inefficient. We propose an alternative approach using a hierarchical structure, which, in the optimal case, results in an exponential increase in the effective rank. We now turn to the hierarchical decomposition of $K$ and $V$:

After hierarchical decomposition, we have:

$$L^{(i)} = \hat{\mathrm{F}}^{(i)}(Q, K)W_S \mathrm{F}^{(i)}(K)^\top, \quad L^{(i)}V^{(i)}$$

For simplicity, let us first consider the case where we do not use hierarchical decomposition during the recursive compression of $K$, and ignore the impact of $\mathrm{Softmax}((QK^\top \odot M))$ for now. The error can be written as:

$$\epsilon = \mathrm{Softmax}((QK^\top \odot M))V - \left(\mathrm{Softmax}((QK^\top \odot M))\sum_i \left(\hat{\mathrm{F}}^{(i)}(Q, K)W_S \mathrm{F}^{(i)}(K)^\top\right) \odot M\right)V^{(i)}$$

This simplifies to:

$$V - \sum_i \hat{\mathrm{F}}^{(i)}(Q, K)W_S \mathrm{F}^{(i)}(K)^\top V^{(i)} = V - \sum_i V'^{(i)}$$

We begin by analyzing the case where the number of levels in the hierarchy is 2. Suppose $C \in \mathbb{R}^{d \times m}$, then the rank of $L_i$ is $m$.

First, the lower bound of the rank can at least represent a vector of length $2m$ without loss, since $L_i$ has rank $m$. When $V^{(i)} = V$, the resulting matrix is the sum of $L_i$, i.e., $L_0 + L_1$, which can achieve a rank of $2m$.

For the optimal case, where $L_i$ has rank $m$, the matrix $V$ is compressed into $m$ directions: $\text{diag}(l)\bar{L}_i$, where $\bar{L}_i$ is the normalized matrix of row vectors from $L_i$, and $l$ represents the amplitude of each row vector. Thus, $\text{diag}(l)\bar{L}_i V^{(i)}$ can be treated as generating $m$ directions.

Further, combining two levels corresponds to vector addition:

$$v_{i \times m + j} = v_i^{(0)} + v_j^{(1)}$$

When the generated vectors $v_{i \times m + j}$ are unequal, the number of directions generated is $m^2$, corresponding to a vector space of dimension $m^2$. In the general case, $c$-level hierarchical decomposition can generate $m^c$ directions, thus enabling precise expression in those directions, with an upper-bound error if the vectors are outside of those directions.

## 3. Error Analysis in Multi-level Decomposition

We proceed to provide a detailed error analysis to illustrate the advantage of multi-level decomposition. Suppose we use $c$ levels of decomposition. Each level reduces the error introduced by the previous level. The vector $V^{(i)}$ at level $i$ is computed as:

$$V^{(i)} = (V^{(i-1)} - L_{i-1}V^{(i-1)})W_{i-1} = (I - L_{i-1})V^{(i-1)}W_{i-1}$$

Thus, the error introduced at the final level is given by:

$$E = V^{(c+1)} = (I - L_c)V^{(c)}W_c = \prod_{i=0}^{c}(I - L_i)V\prod_{i=0}^{c}W_i$$

We can now analyze the trend of the error in detail. The error consists of two main components:

- **Matrix compression error**: $\prod_{i=0}^{c}(I - L_i)$, which represents the cumulative compression of the original vector $V$ through multiple low-rank projections. This term gradually diminishes the contribution of $V$.

- **Weight matrix transformation error**: $\prod_{i=0}^{c}W_i$, which reflects the impact of re-weighting the vector $V$ through the residuals at each level of decomposition.

The total error norm can be bounded as:

$$\|E\| \leq \left\|\prod_{i=0}^{c}(I - L_i)\right\|\|V\|\left\|\prod_{i=0}^{c}W_i\right\| \leq \prod_{i=0}^{c}\|(I - L_i)\|\|V\|\prod_{i=0}^{c}\|W_i\|$$

**Error Trend Analysis**

We now discuss the trend of the error under certain assumptions. To simplify the analysis, we make the following assumptions about the key matrices:

- **Low-rank matrix compression effect**: We assume that the learnable matrix $L_i$ converges to the optimal low-rank approximation of the identity matrix. According to spectral norm analysis, we have:

$$\epsilon_i = \|I - L_i\| = \max\left\{\sigma_1(I - L_i), \ldots, \sigma_m(I - L_i)\right\}$$

Here, $\sigma_i(I - L_i)$ represents the singular values of the matrix $I - L_i$. In the optimal case, $L_i$ is similar to a matrix with $m$ ones on the diagonal and the rest being zeros, where the largest singular value is 1. However, this is the worst-case scenario. As the number of levels increases, the probability of this situation decreases, and thus the expected error becomes more representative.

The expected value of $\epsilon_i$ is:

$$E[\|I - L_i\|] \approx \frac{n - m}{n}$$

We then analyze the effect of the weight matrices $W_i$:

$$\|W_i\| \leq \gamma_i$$

Since $W_i$ are learnable parameters, they tend to converge to eigenvalues less than 1, and thus the error will shrink as $\gamma_i$ decreases.

The overall error norm expectation can be recursively bounded as:

$$\|E\| \leq \|V\| \prod_{i=0}^{c} \epsilon_i \prod_{i=0}^{c} \gamma_i$$

**Convergence Trend Discussion**

The cumulative compression effect: The low-rank compression of the matrix $I - L_i$ gradually suppresses the energy of $V$, and this suppression is multiplicative. When $\epsilon_i < 1$ and does not increase with $i$, the error is exponentially decaying.

In particular, when each level is set to a uniform size such that $\epsilon_i \approx \epsilon$ (a constant value), we have:

$$\prod_{i=0}^{c} \epsilon_i = \epsilon^{c+1}$$

This shows that as the number of levels $c$ increases, the error decays exponentially.

The effect of shrinking the weight matrices: The accumulation of weight matrices $W_i$ reduces the error. To ensure convergence of the error, $\gamma_i$ will become smaller through gradient learning, thereby further reducing the error.

Overall, the error norm converges as:

$$\|E\| \leq \|V\| \prod_{i=0}^{c} (\epsilon_i \gamma_i)$$

If $\epsilon_i \gamma_i$ has an upper bound $\rho < 1$, the error will decay exponentially.

**Numerical Upper Bound and Convergence Trend**

If the values of $\epsilon_i$ and $\gamma_i$ are bounded by fixed upper bounds $\epsilon$ and $\gamma$, respectively, the error convergence rate can be expressed as:

$$\|E\| \leq \|V\| (\epsilon \gamma)^{c+1}.$$

On the other hand, if $\epsilon_i$ and $\gamma_i$ decrease gradually with $i$ (e.g., through optimization of the low-rank matrix $L_i$ and the weight matrices $W_i$), the error decay may be faster than exponential decay. For instance, if $\epsilon_i = \frac{\epsilon_0}{i+1}$, then:

$$\prod_{i=0}^{c} \epsilon_i = \frac{\epsilon_0^{c+1}}{(c+1)!}.$$

This represents a super-exponential convergence.

**Key Conclusions**

1. **Error Trend:** The error is dominated by the cumulative compression factor $\prod_{i=0}^{c}(I - L_i)$, which exhibits exponential decay or faster super-exponential decay, depending on the decay behavior of $\epsilon_i$ and $\gamma_i$.

2. **Optimization Directions:**

   - When selecting $L_i$, it is important to ensure that $\|I - L_i\|$ remains sufficiently small, ideally decreasing gradually.
   - When choosing $W_i$, it is essential to control $\|W_i\|$ so that it does not exceed a certain range.

3. **Practical Implications:** Through careful design of $L_i$ and $W_i$, one can maintain compression efficiency while ensuring that the error converges rapidly. The trend of error reduction across levels becomes increasingly more pronounced.

$\square$

## D. Unification of linear models and their convergence

### D.1. Unified Representations of Attention Mechanisms

We propose a unified formulation of various attention mechanisms. Below, we describe several key models in detail.

**MetaLA:**

$$\text{MetaLA} = \text{MVA}(q_t, k_t^{(0)}, v_t^{(0)}, C^{(0)}, 0, 0, I), \tag{31}$$

$$o_t = \text{GLA}(\phi_q(q_t), (1 - f_\alpha^{(0)}(k_t^{(0)})), v_t^{(0)}, f_g^{(0)}(k_t^{(0)})), \tag{32}$$

where $f_\alpha^{(0)}$ and $f_g^{(0)}$ represent adjustable functions dependent on $k_t^{(0)}$.

**GLA:**

$$\text{GLA} = \text{MVA}(q_t, k_t^{(0)}, v_t^{(0)}, C^{(0)}, 0, 0, I, f_\alpha = 1 - x), \tag{33}$$

$$o_t = \text{GLA}(\phi_q(q_t), k_t^{(0)}, v_t^{(0)}, f^{(0)}(k_t^{(0)})). \tag{34}$$

**LA:**

$$\text{LA} = \text{MVA}(q_t, k_t^{(0)}, v_t^{(0)}, -\infty, 0, 0, I, f_\alpha = 1 - \phi_k(x)), \tag{35}$$

$$o_t = \text{GLA}(\phi_q(q_t), \phi_k(k_t^{(0)}), v_t^{(0)}, I). \tag{36}$$

**GSA:** GSA extends to higher-order interactions by introducing intermediate query gating mechanisms:

$$\text{GSA} = \text{MVA}(q_t, k_t^{(0)}, v_t^{(0)}, C^{(0)}, W_S, 0, 0), \tag{37}$$

$$q_{ks_t} = \text{GLA}(q_t, k_t^{(0)}, (1 - f_\alpha^{(0)}(k_t^{(0)})), f_g^{(0)}(k_t^{(0)})), \tag{38}$$

$$q_{gm_t} = \text{Sigmoid}(q_{ks_t})W_S \approx \text{Softmax}(q_{ks_t}), \tag{39}$$

$$o_t = \text{GLA}(q_{gm_t}, (1 - f_\alpha^{(0)}(k_t^{(0)})), v_t^{(0)}, f_g^{(0)}(k_t^{(0)})). \tag{40}$$

**ABC:**

$$\text{ABC} = \text{MVA}(q_t, k_t^{(0)}, v_t^{(0)}, -\infty, W_S, 0, 0, f_\alpha = 1 - \text{Softmax}(Cx)), \tag{41}$$

$$q_{ks_t} = \text{GLA}(q_t, k_t^{(0)}, (1 - f_\alpha^{(0)}(k_t^{(0)})), I), \tag{42}$$

$$q_{gm_t} = \text{Softmax}(q_{ks_t}), \tag{43}$$

$$o_t = \text{GLA}(q_{gm_t}, (1 - f_\alpha^{(0)}(k_t^{(0)})), v_t^{(0)}, I). \tag{44}$$

## D.2. Delta-like Rule Approximation

Our approach leverages an approximate Delta Rule to maintain state information and minimize information loss during compression. The core idea is to preserve and update states iteratively based on discrepancies between compressed and original sequences. Specifically, we define:

$$k^{(i)}, v^{(i)}, q^{(i)} = W_k x^{(i)}, W_v x^{(i)}, W_q x^{(i)}, \tag{45}$$

$$\hat{v}^{(i)} = W^{(i-1)} \phi(k^{(i)}), \tag{46}$$

$$\beta^{(i)} = \sigma(W_\beta x^{(i)}), \tag{47}$$

$$v_{\text{new}}^{(i)} = \beta^{(i)} v^{(i)} + (1 - \beta^{(i)}) \hat{v}^{(i)}, \tag{48}$$

$$W^{(i)} = W^{(i-1)} + \beta^{(i)} (v^{(i)} - \hat{v}^{(i)}) \otimes \phi(k^{(i)}). \tag{49}$$

To optimize for efficiency, we maintain auxiliary states, such as:

$$W'^{(i)} = \sum_{j=1}^{i} v_j k_j^\top, \tag{50}$$

instead of directly relying on $W^{(i-1)}$. Although this prevents incremental querying of previous states, it introduces performance trade-offs. Our method focuses on the current state and its discrepancy, effectively compressing errors while enabling hierarchical aggregation for improved parallelism and scalability.

