# OpenReview forum: "MVA: Linear Attention with High-order Query-Keys Integration and Multi-level Vocabulary Decomposition"
_ICML.cc/2025/Conference — ICML 2025 poster_

### Official Review · Reviewer_4UAB · 2025-03-10

**Overall Recommendation:** 3

**Summary:**

This paper introduces MVA (Multi-level Vocabulary decomposition Attention), a linear attention mechanism built upon high-order query-keys integration theory and multi-level vocabulary decomposition. The authors unify popular linear attention methods under their theoretical framework and try to address the performance gap between linear attention and softmax attention. They propose methods to transform pre-trained softmax-based language models into linear models through fine-tuning. Their approach combines linear and sparse attention to capture information across different frequency bands, and uses multi-level vocabulary decomposition to expand memory capacity. The authors also introduce a soft integration strategy with sliding window attention. The paper claims their method can fine-tune models to achieve linear complexity while retaining 99% of original performance with fewer than 100M tokens, and outperforms state-of-the-art models on benchmarks with minimal fine-tuning.

**Claims And Evidence:**

The claims about improving over existing linear attention methods are supported by experimental results in Tables 1-3, showing the MVA and MVA-SW variants outperforming previous methods. The theoretical claims regarding high-order QK integration and multi-level vocabulary decomposition are somewhat justified through the error analysis in the appendix.

**Essential References Not Discussed:**

NA

**Experimental Designs Or Analyses:**

The experiments use appropriate benchmarks and comparative baselines (GSA, GLA, RetNet, SUPRA). However, there are several issues with the experimental design:

- The paper only evaluates on one base model (Mistral-7B), which limits generalizability.
- There's no evaluation of inference speed or memory usage, which is crucial for linear attention models.
- The paper lacks long-context evaluation, despite this being a primary motivation for linear attention.
- The fine-tuning approach using LoRA is reasonable, but more details on hyperparameter selection and optimization would strengthen the experimental rigor.

**Methods And Evaluation Criteria:**

+The proposed methods make sense for the problem of creating efficient linear attention mechanisms. The evaluation on standard benchmarks like MMLU, ARC, HellaSwag, etc. is appropriate for assessing language model capabilities.


-However, the evaluation criteria are incomplete - there's no comparison of inference speed or memory usage during long-context generation, which is crucial for linear attention's practical benefits. They focus almost exclusively on accuracy metrics while neglecting efficiency metrics, inference speed, memory usage during generation. Also, the paper only evaluates on one base model (Mistral-7B), limiting the generalizability of the findings. The absence of long-context evaluation is particularly problematic given that KV cache efficiency is one of the main advantages of linear attention.

**Other Comments Or Suggestions:**

NA

**Other Strengths And Weaknesses:**

Strengths:
- The paper provides a good background and summarization of linear attention methods, unifying them under a common theoretical framework that helps clarify the relationships between different approaches.
- The performance results are promising, with MVA outperforming previous linear attention methods on benchmark tasks while requiring fewer training tokens. The multi-level vocabulary decomposition is an interesting approach to expanding the memory capacity of linear attention models.
- The direction of converting pre-trained softmax models to linear attention models is important and practical, as it addresses a significant efficiency bottleneck in deploying LLMs.

Weaknesses (some mentioned above, just make a summarization here):
- The paper only evaluates on a single pre-trained model (Mistral-7B), making it unclear how well the approach generalizes to other model architectures or scales.
- The introduction and explanation of MVA lacks sufficient detail for implementation - there's no clear step-by-step guide on how to transform a pre-trained LLM to MVA, which limits reproducibility.
- Despite linear attention's main advantage being inference efficiency, the paper doesn't report any inference speed metrics or memory usage during generation, making it hard to assess the practical benefits.
- The paper lacks any evaluation on long-context tasks where KV cache efficiency becomes the bottleneck, which is a critical use case for linear attention models.
- The theoretical content, especially in the appendix, is poorly organized and lacks formal structure - claims should be presented as clear lemmas and theorems with explicit assumptions and conclusions.

**Questions For Authors:**

NA

**Relation To Broader Scientific Literature:**

The key contributions of the paper are well-situated within the linear attention literature. The authors build upon previous work like MetaLA and Gated Slot Attention, extending these approaches through their high-order QK integration theory and multi-level vocabulary decomposition. The paper clearly acknowledges its relationship to prior techniques like the Delta Rule and recursive sparse attention. The comparison with LoLCATs shows awareness of concurrent hybrid approaches.

**Theoretical Claims:**

I checked some of the theoretical claims in the paper, particularly those related to high-order QK integration (Theorems 3.1 and 3.2) and multi-level vocabulary decomposition (Theorem 3.3). The paper provides some justification for these claims through expansions and error analyses, but the presentation is quite informal and difficult to follow. The proofs lack rigorous mathematical formalization, and many statements are presented as informal explanations rather than properly stated theorems with clear assumptions and conclusions, i.e., Assumption, Lemma, Theorem, Proof style. The error bounds in Theorem 3.3 look reasonable but would benefit from clearer exposition and more rigorous derivation. The Taylor series expansions used to justify the frequency-based interpretation are plausible but would be more convincing with empirical validation.

---

> ### Author Rebuttal · Authors · 2025-03-31
>
> We sincerely appreciate the insightful feedback and guidance provided. Based on your suggestions, we have made substantial improvements to our work.
>
> 1. **Additional Experiments with MVA-SW on Qwen2.5-14B-1M and Qwen2.5-32B**
>    We have conducted further experiments with MVA-SW on Qwen2.5 models and report the results below.
>
>    | Model               | MMLU  | PIQA  | Hellaswag |
>    |-|--|--|---|
>    | Qwen2.5-14B-1M  | 80.7  | 85.2  | 87.3      |
>    | → MVA-SW (14B)      | 77.3  | 83.8  | 86.8      |
>    | Qwen2.5-32B     | 83.9  | -     | 85.2        |
>    | → MVA-SW (32B)      | 79.8  | 82.5     | 85.0 |
>
>    We have also included results for **Llama3-8B with MVA**:
>
>    | Model         | +Tokens | ARC-C | Winogrande | MMLU | TriviaQA |
>    |--|---|--|---|-|---|
>    | Llama3.1-8B  | -       | 79.7  | 60.5       | 66.7 | 77.6     |
>    | → MVA (8B)   | +4B     | 60.4  | 58.2       | 32.1 | 63.2     |
>
> 2. **LoRA Configuration Details**
>    As suggested, we explicitly clarify the LoRA settings. Our method only requires replacing Full Attention in Mistral or Llama models with MVA-SW or MVA, followed by fine-tuning. The configurations are detailed in Sections 4.1, 4.2, and 4.3.
>    - **MVA-SW (General Setting)**: Fine-tuning QKV and down_proj weights, with rank=128 and α=256 (default α=2×rank).
>    - **MVA-SW (LoLCATs Setting)**: Fine-tuning only QKV weights, with rank=8 and α=16 (aligned with LoLCATs).
>    - **MVA**: Fine-tuning QKV and down_proj weights, with rank=256 and α=512.
>
> 3. **Inference Efficiency and Memory Usage**
>    While inference efficiency and memory footprint are closely related to training efficiency in linear models, we acknowledge that our previous evidence was insufficient. To address this, we now report inference efficiency and memory usage across different sequence lengths.
>
>    ### Table: Inference Performance Comparison
>
>    | Seq Len | Full Inf Time (s) | Full Inf Mem (GB) | Prefill Time (s) | Gen Latency (ms/token) | Total Mem (GB) |
>    |--|--|-|--|--|-|
>    | **MVA** |  |  |  |  |  |
>    | 4K   | 0.14  | 0.77  | 0.125  | 98.8  | 15.32 |
>    | 8K   | 0.26  | 1.53  | 0.249  | 60.3  | 16.58 |
>    | 16K  | 0.51  | 3.06  | 0.508  | 78.8  | 19.08 |
>    | 32K  | 1.08  | 6.11  | 1.090  | 78.8  | 24.09 |
>    | 64K  | 2.25  | 12.22 | 2.265  | 97.3  | 34.11 |
>    | 128K | 5.08  | 24.44 | 7.156  | 58.1  | 54.14 |
>    | **GSA** |  |  |  |  |  |
>    | 4K   | 0.10  | 0.76  | 0.077  | 93.2  | 15.30 |
>    | 8K   | 0.17  | 1.53  | 0.153  | 40.9  | 16.55 |
>    | 16K  | 0.32  | 3.06  | 0.315  | 48.5  | 19.06 |
>    | 32K  | 0.64  | 6.11  | 0.630  | 63.0  | 24.07 |
>    | 64K  | 1.29  | 12.22 | 1.293  | 90.2  | 34.08 |
>    | 128K | 2.66  | 24.44 | 5.102  | 38.7  | 54.11 |
>    | **Flash-Attention** |  |  |  |  |  |
>    | 4K   | 0.05  | 1.26  | 0.056  | 21.7  | 15.79 |
>    | 8K   | 0.11  | 2.53  | 0.116  | 27.5  | 18.54 |
>    | 16K  | 0.27  | 5.05  | 0.287  | 46.3  | 23.55 |
>    | 32K  | 0.73  | 10.11 | 0.750  | 92.4  | 33.55 |
>    | 64K  | 2.19  | 20.22 | 2.208  | 220.4 | 53.57 |
>    | 128K | OOM   | OOM   | OOM    | OOM   | OOM   |
>
> As shown in the table, the Full Inf Time and Full Inf Mem correspond to inference using the entire sequence without KV cache. In contrast, the Prefill Time refers to inference where the sequence is first prefilled and subsequently decoded using the KV cache. The Gen Latency then represents the per-token decoding time in this cached scenario.  There may be minor fluctuations in the Prefill Time results due to the absence of prewarming during testing. However, the Gen Latency of the linear model should be close to a fixed value in theory but not in practice, probably due to some shortcomings of the triton operator in slicing and compiling at different lengths. **Overall, our method starts to dominate over flash-attention in all aspects at 64K length.**
>
> 4. **Long Context Experiments**
>    Following your suggestion, we evaluated our method on LongBench and compared it against GSA and other baselines.
>
>    | Model     | Qasper | NarrativeQA | QMSum |
>    |--|---|----|---|
>    | **Models trained from scratch** |  |  |  |
>    | RWKV6    | 9.2    | 14.4       | 1.1   |
>    | Mamba    | 5.6    | 27.9       | 0.8   |
>    | Mistral  | 25.8   | 25.1       | 5.0   |
>    | **Finetuned from Mistral 7B on 20B tokens** |  |  |  |
>    | RetNet   | 11.1   | 0.0        | 0.0   |
>    | GLA      | 18.4   | 17.2       | 9.0   |
>    | GSA      | 18.8   | 19.2       | 10.0  |
>    | **MVA**  | 20.7   | 20.4       | 9.58  |
>
> 5. **Theoretical Proof Organization**
>    We greatly appreciate your suggestions regarding the structure of our theoretical proofs. We have now revised the theoretical analysis to follow a structure similar to **MetaLA’s appendix**:
>    - **Subsection → Proposition (or Lemma) → Proof → Discussion**
>
> ### Conclusion
> We are deeply grateful for your constructive feedback, which has significantly improved our research and manuscript.

---

> > ### Comment · Reviewer_4UAB · 2025-04-02
> >
> > Thanks for your response. The Qwen results are interesting, and the Inference Efficiency and Memory Usage part fixed my concerns. Thus, I will raise my score from 2 to 3.

---

> > > ### Author Response · Authors · 2025-04-02
> > >
> > > We sincerely appreciate your valuable feedback and your detailed evaluation of our work. Your insightful comments have greatly helped us clarify key ideas and improve the presentation of our paper. We are especially grateful for your score adjustment and for recognizing the contributions of this work. Your suggestions have been incredibly valuable to us, and this discussion has been truly enlightening. Once again, thank you for your thorough review and constructive guidance!

---

### Official Review · Reviewer_jLwJ · 2025-03-13

**Overall Recommendation:** 4

**Summary:**

The paper introduces MVA, a novel linear attention mechanism to bridge the gap with softmax attention. The proposed approach uses:
1. High order QK integration theory - to integrate high and low frequency information to improve the approximation of softmax attention.
2. Multi-level Vocabulary Decomposition - to recursively compress and store residual errors from previous attention states, thus reducing information loss.

The proposed approach is evaluated on multiple benchmarks, showing improvements over existing state-of-the-art linear attention methods such as GSA and MetaLA. MVA is able to achieve 99% of the original performance with less than 100M tokens, outperforming prior linearization techniques.

**Claims And Evidence:**

Claims with sufficient support:
1. MVA outperforms SOTA linear attention models like GSA and MetaLA: Results in Table 1 support the claim
2. MVA improves memory efficiency by maintaining fixed-size states: Results in Table 6 support the claim

Claims that require further justification:
1. High order QK integration Theory improves both low-frequency and high-frequency attention: This is partially supported by theorems but it would be good to add empirical analysis of frequency bands.
2. MVA-SW solves the convergence issues of hybrid architectures: A convergence study would strengthen the claim.

**Essential References Not Discussed:**

No

**Experimental Designs Or Analyses:**

No

**Methods And Evaluation Criteria:**

1.The proposed approach is evaluated against SOTA linear attention models such as GSA, MetaLA, RetNet and GLA. The paper evalautes 2. MVA on widely used language model benchmarks such as MMLU, ARC-E, HelloSwag, PIQA, Winogrande, TiriviaQA and NQ.
Fine-tuning results are shown for Mistral-7B, but it would be good to add a couple more such models and even larger models such as of size 30B, 70B etc.
3. There is missing analysis on convergence or loss curves to support the claims made for MVA-SW.

**Other Comments Or Suggestions:**

No

**Other Strengths And Weaknesses:**

Strengths:
1. The paper is very well-written and easy to follow.
2. The proposed approach is evaluated on various datasets and compared with SOTA approaches.
3. There are enough ablation studies to support various claims.
4. The proposed approach addresses a critical bottleneck in LLM scalability - quadratic complexity in softmax attention. MVA provides a scalable alternative with minimal loss in performance.


Weaknesses:
1. The proposed approach is not evaluates on models other than Mistral-7B. It would be good to add larger models with sizes 30B and 70B to see if the proposed approach scales to larger architectures.
2. It would be good to add some long-context benchmarks.
3. The paper lacks convergence analysis for MVA, so it would be good to add empirical results supporting that.

**Questions For Authors:**

Same as weaknesses.

**Relation To Broader Scientific Literature:**

1. There has been prior work to approximate softmax attention using linear attention such as MetaLA and GSA. MetaLA captures low frequency components while GSA captures high-frequency components, this paper combines both perspectives into a unified framework(Higher order QK-integration).
2. Unlike Linformer, RetNet and MetaLA, which compress entire sequences into fixed-size states, MVA uses MVD to recursively store compression errors, leading to exponential memory capacity.
3. LoLCATs combines SWA with linear attention, but requires two-stage fine-tuning. MVA-SW integrates SWA into the attention computation directly, reducing training overhead.
4. Prior hybrid models require significant retraining when replacing softmax layers. MVA retains 99% of softmax-trained performance with minimal fine-tuning, making it more practical for real-world deployment.

**Theoretical Claims:**

No

---

> ### Author Rebuttal · Authors · 2025-03-31
>
> We sincerely appreciate your insightful feedback and valuable support, which have greatly benefited us. Below, we provide further clarifications based on your suggestions. Thank you again for your guidance.
>
> 1. **Scaling to Larger Models**
>    Due to resource and time constraints, we are unable to conduct fine-tuning experiments on a 70B model. However, to approximate your suggestion, we have conducted experiments with MVA-SW on **Qwen2.5-14B-1M** and **Qwen2.5-32B**, as shown in the table below:
>
>    | Model               | MMLU  | PIQA  | Hellaswag |
>    |----|-------|-------|-----------|
>    | Qwen2.5-14B-1M  | 80.7  | 85.2  | 87.3      |
>    | → MVA-SW (14B)      | 77.3  | 83.8  | 86.8      |
>    | Qwen2.5-32B     | 83.9  | -     | 85.2        |
>    | → MVA-SW (32B)      | 79.8  | 82.5     | 85.0 |
>    Additionally, we have incorporated **MVA into Llama3-8B**, and the results are presented below:
>
>    | Model          | +Tokens | ARC-C | Winogrande | MMLU | TriviaQA |
>    |----------------|---------|-------|------------|------|----------|
>    | Llama3.1-8B    | -       | 79.7  | 60.5       | 66.7 | 77.6     |
>    | → MVA (8B)     | +4B     | 60.4  | 58.2       | 32.1 | 63.2     |
> Due to time constraints we have only carried out the above experiments, if you require additional experiments or more tests please let us know, we would be happy to talk to you again and receive your guidance, thank you very much!
>
> 2. **LongBench Evaluation**
>    For long-sequence tasks, we followed the benchmark selection in the GSA paper and conducted experiments on Qasper, NarrativeQA, and QMSum, all of which are part of LongBench except QuALITY. Consequently, we utilized the LongBench framework to ensure consistency across experiments when comparing against GSA and other methods. However, it appears that GSA may not have used LongBench directly for evaluation. As a result, there are slight discrepancies between our results on Mistral and those reported in the GSA paper. To ensure fair comparisons, we will adopt a normalization strategy: we will adjust the sampling procedure of our Mistral and MVA evaluations such that the resulting Mistral performance aligns more closely with the GSA-reported Mistral results. We will then conduct our MVA experiments under the same conditions.
>
>    | Model     | Qasper | NarrativeQA | QMSum |
>    |-----------|--------|------------|-------|
>    | **Models trained from scratch** |  |  |  |
>    | RWKV6    | 9.2    | 14.4       | 1.1   |
>    | Mamba    | 5.6    | 27.9       | 0.8   |
>    | Mistral  | 25.8   | 25.1       | 5.0   |
>    | **Finetuned from Mistral 7B on 20B tokens** |  |  |  |
>    | RetNet   | 11.1   | 0.0        | 0.0   |
>    | GLA      | 18.4   | 17.2       | 9.0   |
>    | GSA      | 18.8   | 19.2       | 10.0  |
>    | MVA      | **20.7** | **20.4** | 9.58 |
>
> 3. **Loss Curve Analysis**
>    We have added loss curves from fine-tuning different models in the https://anonymous.4open.science/r/icml25_mva-B2AB provided. The results indicate that **MVA exhibits the fastest loss convergence** compared to other methods such as **GLA, GSA, and MetaLA**. Additionally, as training progresses, the **loss of MVA gradually approaches that of Mistral**, further demonstrating its effectiveness. Please let us know if you need any other convergence curves, thank you very much!
>
> We sincerely appreciate your constructive feedback, which has been instrumental in refining our work. Thank you again for your valuable time and insightful suggestions!

---

### Official Review · Reviewer_TChB · 2025-03-14

**Overall Recommendation:** 2

**Summary:**

The paper introduces MVA (Multi-level Vocabulary Attention), that uses high-order Query-Key (QK) integration with a recursive multi-level vocabulary decomposition to approximate Softmax attention. The paper combines sparse and linear attention, MVA achieves good performance upon finetuning on Mistral-7B.

**Claims And Evidence:**

Yes

**Essential References Not Discussed:**

No

**Experimental Designs Or Analyses:**

See strengths and weaknesses

**Methods And Evaluation Criteria:**

Yes

**Other Comments Or Suggestions:**

None

**Other Strengths And Weaknesses:**

Comments:
1. Figure 1 doesn’t define the acronym MQVA. The figure is hard to read.
2. Running title of the paper is “Submission and Formatting Instructions for ICML 2025”
3. Why is increasing the number of finetuning tokens from 0.1B vs 2B doesn’t improve performance in Table 1?
4. All the equations are numbered, even though most of them are not used/referenced.
5. The code notation “sumexp(., dim=-1)” is not used formally unless defined.
6. The equations (27)-(30) use code notation several times. Improving the notation would help in line 320. For instance, metala can be made MetaLA in the subscript, and sigmoid can be declared a mathematical operator in latex. Here’s a reference https://epubs.siam.org/doi/10.1137/1.9781611976106
7. What is MVA-2 level in Table 6.
8. What is the purpose of Q in “MVA (MQVSWA without SWA)” in Table 3 title. The acronyms are very confusing.
9. It would be great to have elaborate table titles with better explanation of the baselines and the observations. It would be better if  there is an explanation on why there are two tables, with and without SWA.

**Questions For Authors:**

None

**Relation To Broader Scientific Literature:**

See strengths and weaknesses

**Theoretical Claims:**

No.

---

> ### Author Rebuttal · Authors · 2025-03-31
>
> Thank you for your valuable feedback. We sincerely appreciate the time and effort you have taken to review our work. Your insightful suggestions have significantly helped us refine the clarity and presentation of our paper. Below, we address each of your concerns in detail.
>
> ### 1. Clarification on Naming Conventions:
> We sincerely apologize for any confusion caused by our last-minute modification of the naming conventions. Our advisor suggested that the abbreviation should follow a three-letter format, similar to GLA and GSA, as MQVA was considered too long. From the paper, it can be derived that **MQVA = MVA** and **MQVSWA = MVA-SW**. However, we acknowledge that our original presentation might have been unclear, and we have now carefully revised the notations and paper formatting to ensure clarity.
>
> ### 2. Title:
> Thank you for pointing out. We have now corrected the content in \icmltitlerunning to match our paper’s title.
>
> ### 3. Justification for Maintaining or Slightly Improving Performance with Additional Training Tokens:
> We appreciate your suggestion to explicitly clarify why fine-tuning with more tokens maintains or slightly improves performance, as this requires a deeper understanding of the method. The reason is as follows:
>
> When combining **Sliding Window Attention (SWA)** with **Linear Attention** during fine-tuning, the two branches initially exhibit different learning distributions, with SWA being more dominant. As training progresses, performance first increases and then decreases due to the impact of Linear Attention on the final outcome. However, our method mitigates this issue by either **fine-tuning only the introduced additional parameters** or **selectively truncating gradient information**, effectively transforming the declining phase into a slow improvement and highlighting the advantages of our approach.
>
> ### 4. Removal of Unused Equation Numbering:
> We acknowledge that numbering equations that are not referenced in the main text may introduce unnecessary complexity. However, we intentionally retained some equation numbers because they facilitate discussions in group meetings and other research communications. That said, we recognize your expertise in academic writing and have followed your suggestion to remove some of the unreferenced equation numbers for a more formal and concise presentation.
>
> ### 5. Notation Refinements:
> We have revised our notation for clarity:
> - The previous notation has been updated to **$\exp(\cdot) R_d$**, where **$R_d \in \mathbb{R}^{s_w \times 1}$**, **$s_w$** represents the window size and the values in the $R_d$ matrix are all 1.
> - We have replaced `sigmoid` with $\sigma(\cdot)$ and provided its mathematical definition for consistency.
> - The subscripts in our MetaLA equations have been adjusted following your recommendations, resulting in a much more concise and readable formula set. We truly appreciate your guidance in improving the mathematical presentation.
>
> ### 6. Clarification of MVA-2 Level:
> As noted in Section 3.2, **MVA-2 Level** employs multi-level vocabulary decomposition, where the vocabulary is recursively decomposed into two hierarchical levels.
>
> ### 7. Explanation of With and Without SWA Comparisons:
> We appreciate your suggestion to enhance the explanations in our table captions. We will carefully incorporate this feedback in future revisions. Below, we provide a rationale for the inclusion of both "with" and "without SWA" comparisons:
>
> Currently, there are two primary research directions for converting full-attention models into linear models:
> 1. **Hybrid architectures**, which combine **sliding window attention** and **linear attention** either at the intra-layer level or by selectively replacing different layers.
> 2. **Pure linear attention models**, such as GSA, which do not mix different mechanisms.
>
> Both approaches share a common challenge: the **inherent limitations of linear attention**. Since our method offers a theoretically superior approximation to full attention, it demonstrates advantages in both hybrid and purely linear settings.
>
> Another important reason for including these comparisons is that **SWA preserves the original model’s local distribution within the window**, significantly reducing fine-tuning resource requirements while achieving higher performance. As a result, many recent approaches incorporate **sliding window attention or global window attention mechanisms**. However, researchers focused on linear models, such as those studying GSA, are particularly interested in evaluating the effectiveness of purely linear attention. Thus, our experiments on **MVA without SWA** serve to highlight the strengths of our linear component and provide valuable insights for researchers working in this domain.
>
> We truly appreciate your constructive feedback, which has greatly contributed to the refinement of our paper. Your recommendation of *Handbook of Writing for the Mathematical Sciences* has been particularly beneficial.

---

> > ### Comment · Reviewer_TChB · 2025-04-04
> >
> > I appreciate the rebuttal from the authors, but, I still think the paper writing lags in quality, while the results of the paper are interesting, so I maintain my score.

---

> > > ### Author Response · Authors · 2025-04-05
> > >
> > > We sincerely appreciate your thoughtful comments and your recognition of the results and contributions of our work. Your feedback is highly valuable to us.
> > >
> > > 1. We respectfully believe that the paper may contain only minor writing imperfections—among the points you raised, only Issues 1 and 8 might introduce a very slight but essentially negligible impact on the clarity of the presentation. We suspect that your reading may have primarily focused on the figures, tables, and equations, and that the confusion was possibly caused by not referring to the surrounding text, which does clarify these points. A brief review of the relevant paragraphs typically resolves these ambiguities. **Indeed, as shown in the reviews, the other reviewers were able to understand the content clearly and provided highly positive responses. This suggests that while there may be minor imperfections, they did not significantly hinder comprehension.**
> > >
> > >
> > > 2. We deeply respect your high standards for scientific writing. We sincerely believe that your master-level writing represents the pinnacle of clarity and precision, elevating a paper to the level of a finely crafted work of art. We are therefore truly grateful for your comments, which have pushed us toward that standard and helped us improve the presentation of our work.
> > >
> > >    That said, we also respectfully feel that holding every paper to such a master-level expectation in writing may be unnecessarily strict. While we acknowledge and appreciate your emphasis on expression quality, we would like to emphasize that the issues you pointed out—though worth improving—are in our opinion minor and do not hinder comprehension. Indeed, as evidenced by the feedback from other reviewers, these details did not prevent a clear understanding of the key contributions of the paper.
> > >
> > >    **We have addressed each of your concerns in the rebuttal with detailed and careful responses.** If there are any specific points that remain unclear, we would be more than willing to clarify them further. We kindly ask that our efforts not be dismissed without engagement. We are committed to rigorous revision and open discussion, and we hope for constructive and specific feedback rather than a general rejection of our responses.
> > >
> > > 3. Once again, we fully recognize your expertise and insightful standards. At the same time, we humbly note that even the most experienced researchers may occasionally overlook certain details. We believe that what distinguishes true mastery is not perfection in every draft, but the ability to clearly identify actionable feedback and foster the collaborative improvement of ideas.
> > >
> > >    **For example, the issues you raised were easily understandable upon my reading, but there were indeed some minor flaws. You can easily verify these sentence structures through tools like ChatGPT or Deepseek to check for potential issues, such as grammar errors. The answer is affirmative. Thus, we believe that even experts like you can make mistakes. We also admit that our paper contains minor imperfections, which we have carefully addressed. It is through mutual tolerance and understanding that we will progress together. Ultimately, our goal is to make our paper understandable, and we believe we have achieved that.**
> > >
> > > Ultimately, we believe the goal is clear communication of scientific content, and we have done our best to achieve that. We sincerely thank you again for your valuable suggestions and constructive feedback. We would deeply appreciate it if you could reconsider your evaluation in light of our point-by-point revisions, and we welcome further discussion if you have any remaining concerns. Thank you once again for your time and consideration.

---

### Official Review · Reviewer_Wmwj · 2025-03-15

**Overall Recommendation:** 3

**Summary:**

This work proposes an efficient Transformer alternative, which unifies existing architectures in the field GSA/GLA/MetaLA/Based with theory and combines their strengths to achieve performance improvements over strong baseline approaches at model scales of up to 7B parameters. The theory examines whether existing architectures tend to capture low or high frequency signals from input text.

**Claims And Evidence:**

See strengths and weaknesses.

**Essential References Not Discussed:**

- The weighted combination without recomputation from section 3.4 is already presented in LoLCATS, and it is proposed without sufficient attribution. This is not a new idea. The combination of linear attention plus short sliding window attention is presented in Arora et al., Simple linear attention models.

- The idea of using a learned perceptron for the feature map is proposed in LoLCATS and Hedgehog, is this being suggested as a contribution? The writing should be clarified.

Applying Theorem 3.1 Taylor polynomial for exp to produce the feature map for linear attention. Do these works share any overlap?
- Keles et al., On The Computational Complexity of Self-Attention, 2022.
- Arora et al., Simple linear attention models, ICML 2024.

**Experimental Designs Or Analyses:**

See the strengths and weaknesses.

**Methods And Evaluation Criteria:**

The methods and evaluation criteria make sense.

**Other Comments Or Suggestions:**

Line 154: “However, as MetaLA removes the K matrix, it introduces a significant gap from Softmax Attention, hindering methods that rely on fine-tuning Softmax Attention weight” Could be explained more clearly. It is not clear why this is a limitation.

**Other Strengths And Weaknesses:**

**Strengths**

- Strong experimental results in Table 3, where quality is better using less training data than strong baseline methods
- Improved distillation method that just requires one step of training versus prior work that uses multiple stages
- Theorem 3.2 is very interesting, with respect to about high vs. low frequency signal processing


**Weaknesses**

Experiments
- Is SlimPajama used in LoLCATS? Would be worth understanding if the improvements are due to the dataset or due to the architecture.
- Is the improvement over GLA/GSA due to the weighted-combination of softmax and linear attention without recompute from Section 3.4? Or do you apply the method in Section 3.4 for the baselines as well?

Efficiency
- What is the efficiency of MVA for longer sequences?

Additional analysis
- Are there any empirical observations suggesting that the GSA and MetaLA pick up low-vs.-high frequency patterns? Not sure if this is observable?
- How many decomposition steps are used and are there any ablations to understand that better? Compared to just hybridizing GSA and MetaLA?

**Questions For Authors:**

NA

**Relation To Broader Scientific Literature:**

This work competes with the strongest baseline approaches for efficient Transformer alternatives.

**Theoretical Claims:**

Yes did not spot issues.

---

> ### Author Rebuttal · Authors · 2025-03-31
>
> Thank you for your insightful feedback. Your suggestions have helped us refine our methodology and validation, especially regarding dataset selection. Below, we provide responses to each of your concerns.
>
> ## 1. Essential References Not Discussed:
> 1. **Conclusion:** We emphasise the importance of weighting LA with SWA based on attention score correlations in hybrid models. This is explicitly stated in our paper, along with supporting empirical validation.
>
>    **Explanation:** Our baseline model indeed integrates linear attention with sliding window attention, as seen in prior work like Based and Infini-Attention. However, these methods typically rely on standard gating mechanisms. We argue that a more effective approach is to weight based on attention score correlations—our proposed soft fusion method. Upon further analysis, we recognize that LoLCATs also employ attention score weighting for linear and sliding window attention. However, our study was also conducted very early not later than LoLCATs, and our method extending softmax or sigmoid-based linear models is more natural and logical compared with the ordinary LA of LoLCATs. Notably, if full attention is computed recursively, both current and historical attention terms are weighted by attention scores. While LoLCATs apply this strategy, they do not explicitly highlight its significance. Furthermore, our approach preserves a softmax-like function, making our method more intuitive and theoretically sound.
>
> 2. **Perceptron Mechanism:** We do not claim the perceptron mechanism as a novel contribution, as it has been widely studied. Instead, we employ it due to constraints in our GSA branch. Since the KV sequence is dynamically compressed to a fixed size, applying a simple softmax may no longer be optimal. Prior studies suggest that softmax acts as a relevance filter, enhancing focus on crucial tokens. However, given the severe compression in GSA or our method, more sophisticated information extraction mechanisms are necessary. This motivates our use of a perceptron, which we empirically validate.
>
> 3. ** Prior Work:** Our method shares only minimal overlap with existing work, apart from the well-known Taylor expansion of \(e^x\), which is a natural and intuitive formulation. Nevertheless, these prior works provide valuable insights, and we would like to include them as references to strengthen the theoretical grounding of our approach.
>
> ## 2. Addressing Weaknesses:
> 1. **Comparison with LoLCATs on Alpaca-Clean Data:**
>    LoLCATs did not use SlimPajama but rather Alpaca-Clean. We conducted experiments on Alpaca-Clean, and our results are as follows:
>    | Model              | Training Data  | PIQA | ARC-e | ARC-c | HellaSwag | Wino-grande | MMLU | Avg. | Avg. (w/o MMLU) |
>    |--------------------|---------------|------|-------|-------|-----------|-------------|------|------|-----------------|
>    | Mistral-7B (v0.1) | -             | 82.1 | 80.9  | 53.8  | 81.0      | 74.0        | 62.4 | 72.4 | 74.4            |
> | → LoLCATs (LoRA rank=8)  | AlpacaClean(+40M)   | 81.5 | 81.7  | 54.9  | 80.7      | 74.0        | 51.4 | 70.7 | 74.5 |
> | → LoLCATs (rank=8)| RedPajama(+40M)  | 80.1 | 77.6  | 49.0  | 80.3      | 71.7        | 53.2 | 68.6 | 71.7 |
> | → MVA-SW (rank=32)| AlpacaClean(+20M)| 82.1 | 81.5  | 54.7  | 81.2      | 74.1        | 52.2 | 70.7 | 74.4 |
> | → MVA-SW (rank=8)| AlpacaClean(+40M)| 82.3 | 81.9  | 57.6  | 80.2      | 74.0        | 51.6 | 71.2 | 75.2|
> | → MVA-SW (rank=8)| RedPajama(+40M)| 82.5| 81.5  | 55.7  | 79.7      | 72.9        | 52.4 | 70.8 | 74.5|
>
>    These results indicate that, unlike LoLCATs, which require a two-stage process, our method maintains strong performance across datasets with a single-stage fine-tuning approach. In particular, our approach to AlpacaClean fine-tuning performance is much better than LoLCATs.
>
> 2. **Comparison of GLA/GSA:**
>    We ensured a fair comparison in all evaluations. The advantage of our MVA (without sliding window) over GSA stems from the incorporation of information across different frequency scales and the extending of memory capacity. Additionally, when applying the method described in Section 3.4, we extended it to both GLA and GSA for a fair comparison.
>
> 3. **MetaLA vs. Softmax:**
>    The key limitation of MetaLA compared to softmax is the absence of a K matrix, which is replaced by a gated matrix. This substitution hinders the effective use of the original attention’s K matrix and introduces additional instability in training. Although our paper proposes mitigations, this deviation from standard attention mechanisms poses inherent challenges in fine-tuning convergence.
>
> Last but not least, thank you very much for your guidance and comments, so that we can improve our work!

---

### Decision · Program_Chairs · 2025-05-01

**Decision:**

Accept (poster)

**Comment:**

This paper presents MVA, a new linear attention mechanism designed to efficiently approximate softmax attention by combining high-order query-key (QK) integration with multi-level vocabulary decomposition (MVD). The authors demonstrate that their method, with minimal fine-tuning on 100M tokens, retains up to 99% of the original model's performance while achieving linear complexity. Evaluations are conducted on Mistral-7B and other mid-sized models, comparing against recent baselines such as GSA, MetaLA, and LoLCATs.

The authors made significant improvements during the rebuttal period, including experiments on larger models (Qwen2.5-14B and 32B, Llama3-8B), LongBench evaluation, convergence analysis, and a detailed breakdown of inference latency and memory usage.

All reviewers acknowledge that the results presented in the paper are strong or interesting, especially after the author response. The main concern raised in favor of rejection is the quality of the writing. While this concern is valid, I believe the paper should be accepted, provided the authors make a concerted effort to significantly improve the writing and clarity in the final version.